# Asymmetric magma plumbing system beneath Axial Seamount based on full waveform inversion of seismic data

Jidong Yang [1] ✉, Hejun Zhu [2,3], Zeyu Zhao[4] ✉, Jianping Huang[1] ✉, David Lumley [2,3], Robert J. Stern [2], Robert A. Dunn [5], Adrien F. Arnulf[6,7] & Jianwei Ma [4]

The architecture of magma plumbing systems plays a fundamental role in volcano eruption and evolution. However, the precise configuration of crustal magma reservoirs and conduits responsible for supplying eruptions are difficult to explore across most active volcanic systems. Consequently, our understanding of their correlation with eruption dynamics is limited. Axial Seamount is an active submarine volcano located along the Juan de Fuca Ridge, with known eruptions in 1998, 2011, and 2015. Here we present high-resolution images of P-wave velocity, attenuation, and estimates of temperature and partial melt beneath the summit of Axial Seamount, derived from multi-parameter full waveform inversion of a 2D multi-channel seismic line. Multiple magma reservoirs, including a newly discovered western magma reservoir, are identified in the upper crust, with the maximum melt fraction of ~15–32% in the upper main magma reservoir (MMR) and lower fractions of 10% to 26% in other satellite reservoirs. In addition, a feeding conduit below the MMR with a melt fraction of ~4–11% and a low-velocity throat beneath the eastern caldera wall connecting the MMR roof with eruptive fissures are imaged. These findings delineate an asymmetric shallow plumbing system beneath Axial Seamount, providing insights into the magma pathways that fed recent eruptions.

The Juan de Fuca Ridge (JdFR) is an oceanic spreading center in the northeast Pacific Ocean with an intermediate spreading rate ~5.6 cm/year[1–3]. The most prominent feature on the JdFR is Axial Seamount (Fig. 1a), an active submarine volcano with a summit rising ~1 km above the adjacent seafloor and a narrow 11-km-deep crustal root[4,5]. Because of enhanced magma supply from the juxtaposition of the Cobb hotspot and the JdFR[6], Axial Seamount is volcanically active, with known eruptions in 1998, 2011, and 2015[7–9]. Intensive studies of this submarine volcano since the 1990s have monitored its activity and revealed aspects of its underlying magmatic system. For instance, a refraction tomography study in 2001 using ocean-bottom seismometers (OBSs) imaged a 250-km³ low velocity zone beneath Axial Seamount[10]. Later, multi-channel seismic (MCS) experiments refined this model and identified two large elongated low-velocity magma reservoirs[5,11]: a 14-km long, 3-km wide, and up-to-1-km thick main magma reservoir (MMR) with a roof ~1.1–2.3 km beneath the summit caldera, and a

[1]National Key Laboratory of Deep Oil and Gas, School of Geosciences, China University of Petroleum (East China), Qingdao, Shandong, China. [2]Department of Sustainable Earth Systems Sciences, The University of Texas at Dallas, Richardson, TX, USA. [3]Department of Physics, The University of Texas at Dallas, Richardson, TX, USA. [4]School of Earth and Space Sciences, Peking University, Beijing, China. [5]Department of Earth Sciences, University of Hawaii, Honolulu, HI, USA. [6]Institute for Geophysics, University of Texas at Austin, Austin, TX, USA. [7]Present address: Amazon Web Services, Seattle, CA, USA.
✉ e-mail: jidong.yang@upc.edu.cn; zy.zhao@pku.edu.cn; jphuang@upc.edu.cn

**Fig. 1 | Geological setting of Axial Seamount on the Juan de Fuca ridge (JdFR) and multi-channel seismic data used in this study. a** Major tectonic plates and their boundaries (green dashed lines) near the JdFR. **b** 3D view of bathymetry near Axial Seamount. The thick black solid line shows the location of the J48 survey line[5]. Yellow polygons denote the extent of main and secondary magma reservoirs (MMR, SMR) identified in ref. 11 and a small western magma reservoir (WMR) identified in this study. The thin dashed black line represents the 2011 and 2015 eruptive fissures. Green, blue, and cyan polygons mark the extent of 1998, 2011, and 2015 lava flows, respectively[7–9,15]. Black dots denote seismicity prior and during 2015 eruption[5]. Magenta dots denote the location of hydrothermal vents. Dashed white lines denote the contours of 10-km and 11-km crust thickness[4]. **c, d** Comparisons of observed (black) and predicted data (red) from the starting tomographic model[5] and final full waveform inversion models, respectively. The bottom (**c, d**) compare peak amplitudes for crustal refractions highlighted by green triangles. Incorporating the attenuation effects produces better waveform fitting for large offsets.

secondary magma reservoir (SMR) beneath the eastern shoulder of the volcano (Fig. 1). In addition, below the southern end of the MMR, a vertical stack of melt sills were imaged between 2.5 and 4.5 km below the seafloor (bsf) within a 3–5-km-wide mush feeder conduit, providing new insights into magma accumulation and transport in the lower crust for active submarine volcanoes[12,13]. Multibeam data collected from autonomous underwater vehicles mapped the eruptive fissures associated with recent eruptions along the eastern caldera rim, as well as the north and south rift zones[14,15]. Long-term geodetic observations of seafloor deformation demonstrated an inflation-predictable eruptive behavior of Axial Seamount, and were used to successfully forecast the 2015 eruption[16,17]. Furthermore, earthquake catalogues show that outward-dipping ring faults beneath the eastern and western caldera walls were reactivated to accommodate MMR inflation and deflation[18–20]. Although previous studies have revealed the location and geometry of the MMR, it remains unclear whether additional finer-scale magma reservoirs exist and how magma conduits are distributed in the subsurface.

Identifying magma storage beneath volcanic edifices is important for understanding volcano dynamics, crustal accretion, and eruption triggering. The OBS refraction tomography study[10] estimated a magma volume ranging from 5 to 21 km$^3$, with melt fractions up to 5–25%. Shear wave velocities constrained from continuous seafloor compliance data were used to calculate a melt fraction of 14% in the shallow magma reservoir and at least 4% in the lower crust beneath the central caldera[21]. These estimates are averages over large magma volumes and therefore suffer from limited resolution and large uncertainties. In addition, they fall below the critical threshold (~32–50%) observed in laboratory experiments for mobilizing magma to erupt from a crystal mush[22–24], which is inconsistent with the observed frequent volcanic activity of Axial Seamount[14,15]. More recent joint tomography using both OBS and MCS data further refines the melt volume as ~27–87 km$^3$ with a maximum melt fraction between 39% and 65%[5], which is more than two orders of magnitude greater than the amount of melt emplaced during the 2015 eruption (total magma volume of ~0.29 km$^3$ including ~0.15 km$^3$ of seafloor lava flows and ~0.14 km$^3$ of subsurface dikes[5,9,15,17]). Limited by the low resolution of seismic tomography study, the melt fractions in the high-melt zone beneath the southeast caldera could not be accurately determined, and it is uncertain whether its volume is comparable to the size of recent magmatic events. Moreover, the reason for the connection between the high-melt zone and the focusing of eruptive fissures near the eastern caldera wall in 1998, 2011, and 2015 remains unclear.

Here, we show high-resolution images of P-wave velocity, attenuation, and estimates of temperature and partial melt beneath the summit of Axial Seamount along JdFR, derived from an advanced multi-parameter full waveform inversion (FWI) method for MCS data. The MMR, SMR, and a small magma reservoir beneath the western

caldera wall are imaged at high resolution. A low-velocity throat beneath the eastern wall, connecting the MMR roof with eruptive fissures on the seafloor, is identified. These findings reveal an asymmetric shallow plumbing system beneath Axial Seamount, shedding light on the magma pathways that fed recent eruptions.

## Results and discussion

### P-wave velocity and attenuation models across Axial Seamount

Twelve MCS profiles were collected in 2002 over Axial Seamount with a 49.16-L airgun source array fired every 37.5 m, and each source was recorded by a 480-channel, 6-km-long Syntron streamer with a 12.5-m hydrophone interval[2,11,25]. Because the sources and receivers of MCS data are located far from the seafloor, oceanic crust refractions are usually masked by strong seafloor scatterings, making it challenging to employ seismic tomographic methods. To enhance the energy of crustal refracted arrivals, MCS data can be downward continued to simulate a synthetic ocean bottom experiment with virtual sources and receivers located on the seafloor, which is easier to handle for tomographic imaging[26,27]. FWI is an advanced seismic tomography method based on full wavefield numerical simulations, which can accurately constrain complex subsurface anomalies of wave speeds, attenuation, and anisotropy[28–31]. In this study, we started from a traveltime tomography model (Fig. S1) and performed a viscoacoustic

FWI (see Methods)[32] for downward continued MCS data to simultaneously constrain both P-wave velocity ($V_p$) and attenuation along the J48 MCS survey line that crosses Axial Seamount from southwest to northeast (Fig. 1b).

The attenuation, which is commonly parametrized by the quality factor ($Q_p$), is a metric for quantifying energy dissipation and phase dispersion of anelastic seismic waves[33]. Because of its strong dependence on temperature, partial melt, and water content, mapping anelastic attenuation can provide valuable information on Earth's internal structure and dynamics, complementing seismic velocity information[34]. P-wave propagation in an anelastic medium can be described by using a viscoacoustic wave equation[35]. A multi-scale inversion strategy was implemented by gradually incorporating bands of increasing frequency (3–5 Hz, 3–7 Hz, 3–9 Hz) to produce improved $V_p$ and $Q_p$ models (Figs. S2–S5). In each frequency band, $V_p$ and $Q_p$ models were first simultaneously updated and later the $Q_p$ model was updated alone, which allowed us to minimize tradeoffs between these two model parameters. The total misfits of the frequency bands are reduced by 22.6%, 67.8%, and 41.3%, respectively (Figs. S6–S9). Crustal refraction waveforms calculated using the final velocity and attenuation models (Fig. 2) fit better with observed data in terms of both traveltimes and amplitudes than the results from the initial models (Fig. 1c, d; Figs. S3–S5). Extensive synthetic tests demonstrate that the

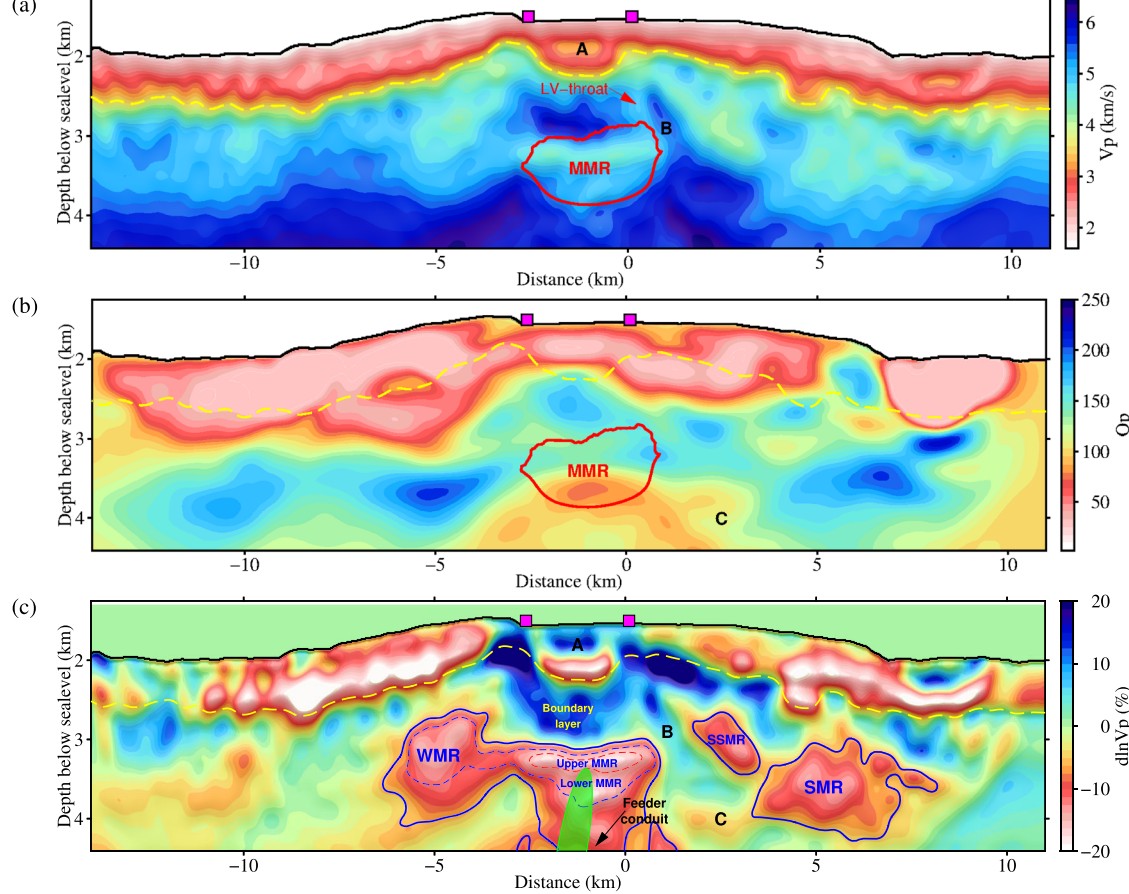

**Fig. 2 | P-wave velocity and attenuation models beneath Axial Seamount.**
**a** P-wave velocity model estimated from viscoacoustic full waveform inversion.
**b** Estimated P-wave attenuation model. **c** Relative velocity anomaly model ($\delta \ln V_p$, see Methods for detailed calculation). Yellow dashed lines denote the bottom of layer 2A. Purple rectangles mark the locations of hydrothermal vents. Green shaded ellipse in (**c**) indicates the location of a revised prolate-spheroid inflation source for the 2015 eruption[17,52]. Red polygons in (**a**) and (**b**) outline the main magma reservoir (MMR) identified by ref. 11, but multiple magma reservoirs are revealed in this study

indicated by solid blue polygons in (**c**). Dashed blue contour in (**c**) outlines the MMR and a western magma reservoir (WMR) and red dashed contours denote the upper MMR. SMR: secondary magma reservoir. SSMR: shallow secondary magma reservoir. Label A highlights a high-velocity core on the subsiding caldera floor, B marks the east-dipping high-velocity belt between the MMR and SMR, and C shows a low-velocity and low-$Q_p$ subhorizontal anomaly connecting the axial magma conduit and the western flank of the SMR.

inverted $V_p$ and $Q$ models are reliable by the FWI method (Methods; Figs. S10–S12). For example, the MMR with a −17% velocity anomaly and a $Q_p$ value as low as 35 at 2.0–2.5 km bsf, as well as shallower magma pockets ranging from 570 m to 1100 m wide and from 150 m to 400 m thick at 0.9–1.5 km bsf, can be accurately imaged by using the same data acquisition and inversion scheme (Fig. S11).

Compared to previous tomographic results[5] (Fig. S1), our models reveal more detailed volcanic structure below Axial Seamount (Fig. 2). For instance, most P-wave velocities are less than 3 km/s in oceanic sediments and the shallow basaltic layer, except for a 1.2-km-wide, 0.2-km-thick, high-velocity anomaly ($V_p \sim 3.2$ km/s) at 246 m below the caldera (Label A in Fig. 2a). This may be attributed to the presence of thick, dense, ponded lava flows on the subsiding caldera floor. This high-velocity core is bounded by narrow inward-dipping low-velocity zones, which are co-located with inward-dipping ring fault zones identified from caldera walls and seismicity distribution[5,20]. The estimated $Q_p$ above the base of layer 2A ranges from 15 to 60 (Fig. 2b), which is consistent with tomographic results for the East Pacific Rise[36] and laboratory measurements of JdFR basalt samples[37–39]. The low-$Q_p$ values indicate strong seismic attenuation within the shallow oceanic crust, which might result from the high porosity of the basaltic extrusive layer[8], faulting and deformation associated with volcano inflations and deflations[18], cracking induced by hydrothermal circulation[36], as well as scattering from velocity heterogeneities[40,41]. Based on a relationship between $Q_p$ and porosity for fractured and altered basalt from ODP site 504B[39], we estimate a maximum porosity of ~10–16% in layer 2A (see Method; Fig. S13). This estimation is consistent with the measured porosity of Oman Ophiolite pillow basalt (4–12%)[42], but it is less than that of the uppermost oceanic crust at 100–200 m depth estimated from gravity surveys (for instance, 10–23% on the Endeavour segment[43], 15–33% for Axial Seamount[44]). This may be caused by the averaging effect of FWI along wave paths over near-seafloor sediments and deeper parts of layer 2A (~287–925 m).

The $V_p$ anomaly pattern (Fig. 2c) reveals the presence of what we interpret to be multiple magma reservoirs in the shallow crust, indicated by blue and red contours plotted according to the estimated partial melt fractions shown in Figs. 3–4. Beneath the high-wavespeed thermal boundary layer[13], a ~3-km-wide and ~250-m-thick slow P-wave speed anomaly is resolved in the upper MMR at ~1.7 km bsf (dashed red polygon in Fig. 2c), with an ~18% velocity reduction. This is consistent with strong amplitudes of coincident seismic reflections, indicating relatively high melt fraction in the upper MMR[11]. A smaller $V_p$ reduction (~12%) is found in the lower MMR (dashed blue contour in Fig. 2c), underlain by a 1.6 to 2-km-wide low-velocity conduit that extends down to the mid-to-lower crust. Stacked melt lenses characterized by bright reflections have been identified in the deep conduit[12], but with a larger width (3–5 km) than observed in our FWI model. In the lower MMR, a low-$Q_p$ anomaly is observed with the lowest value ~60 (Fig. 2b), reflecting strong seismic attenuation and high crystal mush. However, the upper MMR has slightly increased $Q_p$ values (70–100, Fig. S9c), which may be caused by the decreased viscosity in a framework of fluid-suspended crystals with a higher melt fraction close to the eruption threshold[24]. An incomplete correlation between velocity and attenuation structures suggests their different responses to temperature and partial melt[34]. Furthermore, two separated low-velocity zones are identified at the location of the previously recognized secondary magma reservoir, located 3–10 km east of the caldera[5] (Fig. 2a, c): a shallower small one adjacent to the eastern shoulder of the caldera (here named the SSMR), and a deeper larger one (here named the SMR) located ~7 km east of the MMR with a top ~1.2 km bsf. Estimated attenuation in the SMR and SSMR is not as strong as within the MMR, reflecting a lower melt fraction. A narrow (0.7–1.2 km wide) east-dipping high-velocity belt at 1–2 km bsf between the MMR and SMR implies that they are not connected along the J48 survey line (Label B in Fig. 2a, c). However, a low-velocity and low-$Q_p$ subhorizontal anomaly

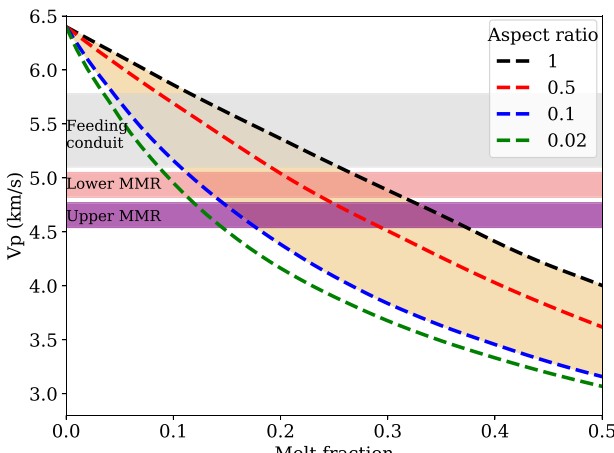

**Fig. 3 | Melt fraction modeling based on two-phase differential effective medium theory**[49]**.** $V_p = 6.4$ km/s, $V_s = 3.5$ km/s and density = 2.97 g/cm³ for solid basalt, and $V_p = 2.35$ km/s, $V_s = 0$ km/s and density = 2.7 g/cm³ for basaltic melt[5]. Black dashed line denotes spherical inclusions (aspect ratio of 1), and red, blue and green dashed lines are for vertically aligned elliptical melt inclusions with aspect ratios of 0.5, 0.1, and 0.02, respectively. Purple, orange, and gray shaded regions indicate the melt extent for the upper main magma reservoir (MMR), lower MMR, and feeding conduit.

at ~2.5 km bsf, connecting the deep magma conduit below the MMR and the western flank of the SMR (Label C in Fig. 2b, c), indicates that the SMR may result from tapping of deep melts in the mushy zone of the deep conduit[13,45].

In addition to the MMR and SMR, two previously unrecognized low-velocity features are mapped in our FWI models. The first is a west-dipping oval pocket beneath the western caldera rim, extending from 1.1 km to 2 km bsf, which appears to connect with the upper MMR through a ~220-m-thick channel (Fig. 2a, c). This western magma reservoir (WMR) exhibits ~12% $V_p$ reduction and low-$Q_p$ below layer 2A and is co-located with strong reflection events to the west-southwest of the MMR[12], indicating that this is a third region with relatively high melt fraction (Fig. S14). The distribution of seismicity in and around the caldera at Axial shows that the west-dipping ring fault beneath the western caldera rim terminates in a notch between the MMR and WMR. This suggests enhanced cooling in the western shoulder and the absence of active magma conduits from the deep high-melt reservoir to the seafloor below the western edge of the caldera. Consistently, few eruptive fissures have been observed in this area for the last several hundred years, as mapped by high-resolution multibeam data[14]. Second, a low-velocity throat is imaged below the eastern caldera wall that connects the MMR roof to the bottom of layer 2A (Fig. 2a). The presence of this throat aligns with the depth extent of seismicity beneath the eastern caldera wall[5,20] and shallow-dipping reflectors between the base of Layer 2A and MMR[11], suggesting a magmatic pathway for recent diking events and eruptions[18].

## Differential partial melts in the magma reservoirs and conduit

Based on the FWI velocity and attenuation models in Fig. 2, we estimate the thermal structure and partial melt fraction beneath Axial Seamount by considering both anharmonic and anelastic effects. The anharmonic thermal model is calculated by adding the temperature anomaly estimated from the $V_p$ anomaly to a one-dimensional oceanic thermal model, while anelasticity is incorporated by iteratively updating the temperature model (see Methods)[46,47]. We try to explain the low-velocity zones as much as possible by thermal anomalies alone below layer 2A, but partial melt is required for temperatures above 1150 °C[48]. Assuming a two-phase effective medium consisting of solid and molten basalt (crystal mush with interstitial partial melt)[5,49], we estimate

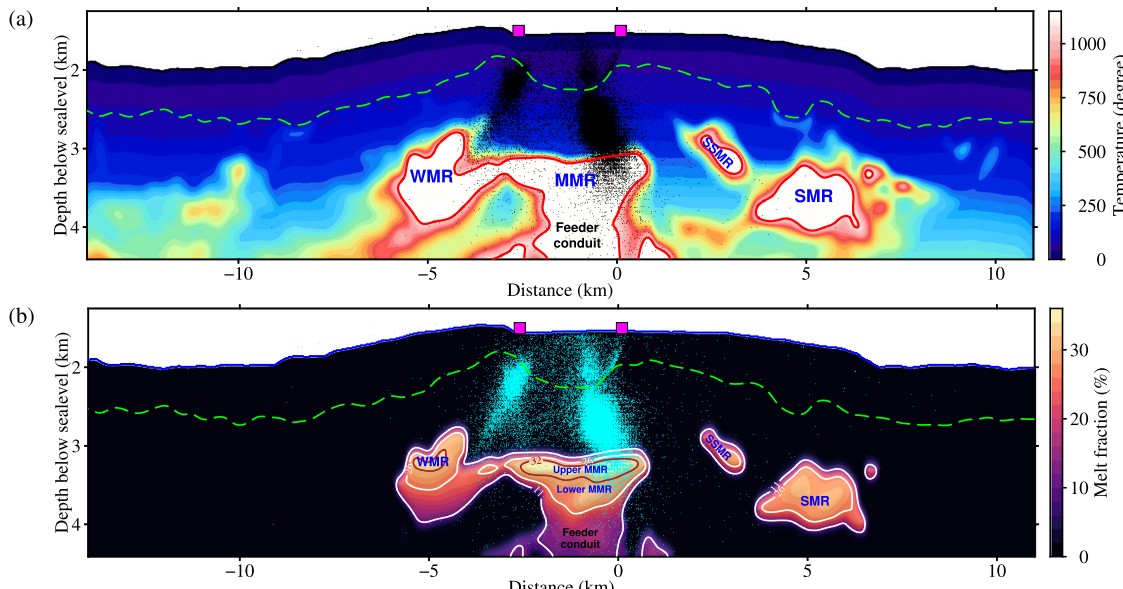

**Fig. 4 | Temperature and melt distributions beneath Axial Seamount. a** A temperature model with a 1150 °C cutoff assuming both anharmonic and anelastic effects[46]. **b** Estimated melt fraction within regions with temperature above 1150 °C assuming spherical inclusions. Green dashed lines show the bottom of layer 2A. Purple rectangles denote the locations of hydrothermal vents. Red solid lines in (**a**) show the temperature outline of the 1150 °C isotherm. Brown contour in (**b**) marks a high-melt zone (~32%), and white contours outline melt zones with ~26% and ~11%. MMR: main magma reservoir. WMR: western magma reservoir. SMR: secondary magma reservoir. SSMR: shallow secondary magma reservoir.

the melt fraction in regions with temperatures greater than 1150 °C (Fig. 3; Methods). Our final thermal and melt models across Axial Seamount are shown in Fig. 4 and Figs. S15–19.

$V_p$ perturbations far from the summit caldera can be explained by thermal effects alone, with temperatures ranging from ~4 °C near the seafloor to ~440 °C at 2.5 km bsf. However, the thermal effect alone is not enough to explain the observed prominent $V_p$ reductions for the MMR, SMR, and WMR, where partial melt is required. Strong seismic attenuation of partial melts makes the anelastic effect significant for constraining the structure of the magma reservoirs and the deeper feeding conduit, resulting in a much narrower region enclosed by the 1150 °C isotherm than from the anharmonic model (Fig. 4a, Fig. S15). Spherical and vertically aligned elliptical melt bodies with an aspect ratio of 0.1 are considered as two end members for estimating the melt fractions[5,50]. The upper MMR has the largest melt fraction, ranging from 15% to 37% (Fig. 4b; Figs. S16–20). We estimate the volume of basaltic melt by first applying an area integral over the melt region based on our melt fraction model and then using the 3D reservoir geometry (Fig. S21) extracted from the traveltime tomography model in ref. 11 to constrain the length. The melt volume in the upper MMR is ~0.27–0.57 km³, which is similar to the volume of lava flows and magma intrusions in the fissures during the most recent three eruptions (1998[8], ~0.21 km³; 2011[16], ~0.15 km³; 2015[17], ~0.29 km³). Considering the eastern seismicity cluster connecting eruption vents to the MMR roof at depth, we infer that the high-melt zone in the upper MMR beneath the southeastern caldera is the direct magmatic source for recent eruption and diking events. In the SMR, WMR, and lower MMR, a slightly smaller melt fraction of ~10–26% is mapped. The feeding conduit in the middle-to-lower crust has a melt ratio of ~4–11%, consistent with the estimates from continuous observations of seafloor compliance (requiring at least 4%)[21]. Based on the melt distribution shown in Fig. 4b and Fig. S18b, we estimate a total magma volume of ~12.38–29.60 km³ in the upper and middle crust. This agrees with those estimated by seismic tomography from OBS and MCS data[5,10], which is about two orders of magnitude greater than the volume emplaced during recent eruptions. The spatially variable melt content forms thermally and chemically zoned magma reservoirs in shallow crust[48,51], which generated seismic reflections with varying amplitudes[11] and was tapped by a dike feeding the 2015 lava flows with a range of MgO content[9].

## An asymmetric magma plumbing system beneath Axial Seamount

The high-resolution P-wave velocity and attenuation models as well as estimated temperature and melt distributions from this study reveal a complex asymmetric magma plumbing system beneath Axial Seamount (Fig. 5). Below a thermally controlled permeability barrier, the MMR formed through magma accumulation and pooling at the base of the sheeted dike layer (Layer 2B). The top of the MMR has a melt fraction up to 37%, approaching the threshold (~35–50%)[24] for magma mobilization and eruption. The estimated volume of this high-melt fraction (~0.27–0.57 km³) is slightly greater than the volume of recent eruptions (~0.15–0.29 km³), suggesting that the top of the MMR is the major magma source for them. Beneath the eastern caldera wall, a low-velocity magma throat (yellow in Fig. 5) parallel to the outward-dipping fault plane connects the roof of the underlying high-melt reservoir to a shallow dike zone (magenta belt in Fig. 5) that feeds eruptive fissures.

On the other hand, beneath the western caldera wall, similar conduits are not observed in the pillow basalt and sheeted dike layers, although a small magma pocket (WMR) is imaged to the west of the MMR. A thin, subhorizontal melt band connects this pocket with the MMR, providing a magma delivery conduit. As the MMR is replenished during inflation and depleted during deflation, the stress imposed on the western caldera wall can be relaxed by magma transport into or out of the WMR, and the remaining stress associated with viscous magma movement is accommodated by slip along the west-dipping ring fault. However, the stress of the eastern wall is only accommodated by reactivating the east-dipping ring fault during volcanic inflation and deflation, leading to magma intrusion along the fault plane and forming a low-velocity diking throat. Relaxation of the WMR produces differential stresses on the two ring faults, which might explain the observation that less seismicity occurs beneath the west wall than the east wall[18–20] (Fig. 4) and that the eastern caldera fault had the most slip during the 2015 eruption[52].

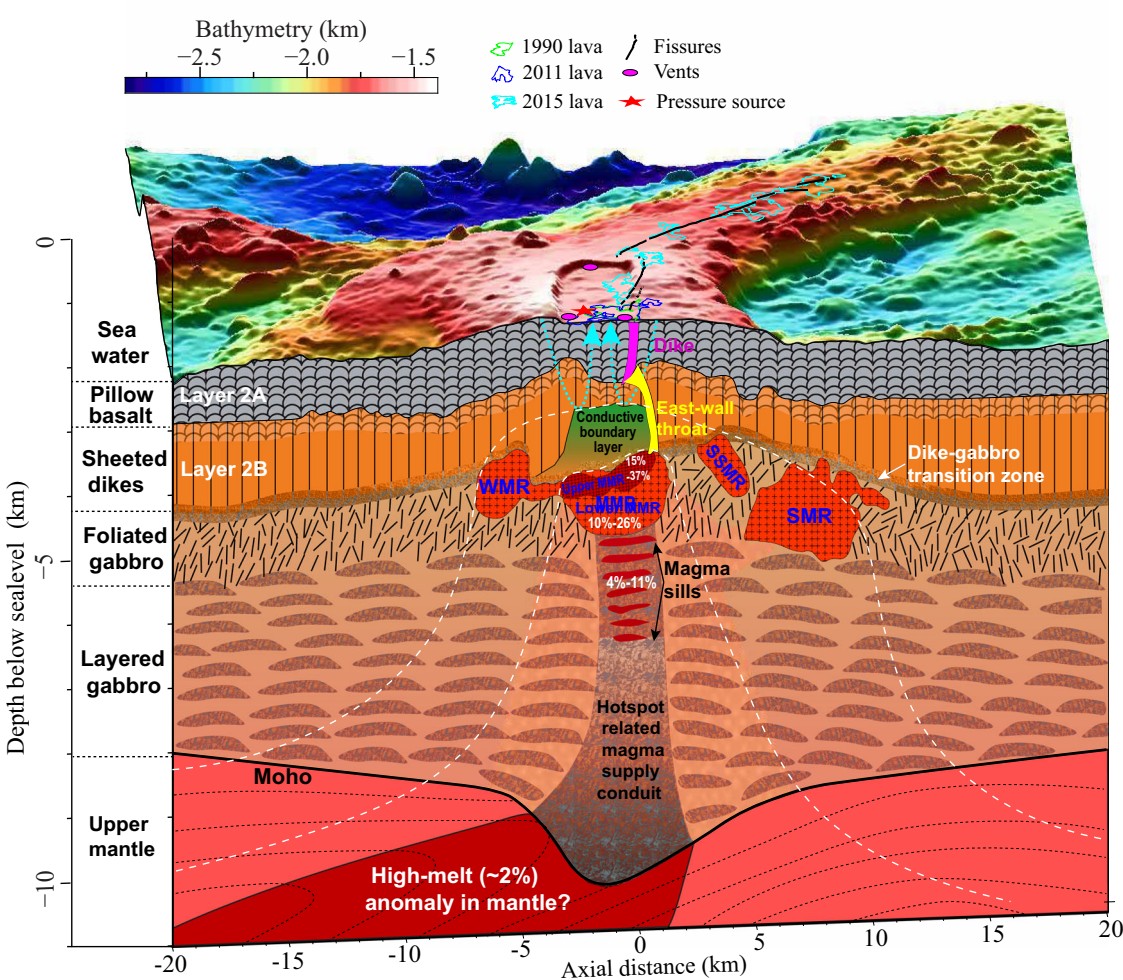

**Fig. 5 | A schematic model of the magmatic system below Axial Seamount.** The geometry of asymmetric magma plumbing and melt fractions in the upper crust is based on full waveform inversion models in Fig. 2 and estimated melt fraction in Fig. 4. The distribution of stacked sills in the mid-to-lower crustal melt-mush feeder conduit is according to seismic reflection images in ref. 12. The Moho depth, west-dipping mantle flow (dashed streamlines) and ~2% melt (dark red) in the upper mantle are based on refs. 4,54,78. Green, blue, and cyan polygons on the seafloor indicate 1998, 2011, and 2015 lava flows, respectively. The red star indicates the location of a revised prolate-spheroid inflation source for the 2015 eruption[52], and the magenta cycles denote the hydrothermal vents. Black lines denote eruptive fissures identified during 2011 and 2015 eruptions. The yellow shaded area illustrates the magma throat beneath the eastern caldera wall that connects the main magma reservoir (MMR) to layer 2A. The magenta shaded area indicates the diking path constrained by seismicity[18]. A thermally controlled permeability boundary layer separates the MMR from hydrothermal fluid circulation (cyan dashed lines with arrows) in the dike and lava section above[13]. White dashed lines denote the isotherms. WMR: western magma reservoir. SMR: secondary magma reservoir. SSMR: shallow secondary magma reservoir.

Continuous geodetic monitoring of the caldera identified eight discrete short-term deflation events between August 2016 and May 2019[53], accompanied with low seismicity rates. During the June 2018 event, the bottom pressure recorder near the western caldera wall experienced slightly greater vertical displacement (subsidence) than those near the eastern wall. This could be consistent with short-distance magma movement from the MMR to the WMR, combined with viscoelastic relaxation around the MMR associated with a temporally interrupted magma supply[53]. The MMR, the WMR and the narrow diking throat on the eastern wall form an asymmetric magmatic plumbing system in the shallow crust (Fig. 5). In this system, the scarcity of observed diking pathways near the western caldera wall and the connection between the high-melt zone of the MMR and seafloor fissures via the eastern magma throat may explain why the recent eruptions have initiated along the eastern edge of the summit caldera.

Beneath Axial Seamount, a narrow-focused magma supply conduit (1.6–2-km-wide, ~4–11% melt fraction) is mapped at the bottom of the MMR and extends to the mid-to-lower crust (Figs. 4, 5). Its location is consistent with the center of the thickened crust root revealed by PmP traveltime tomography[4], a series of vertically stacked melt lens on MCS reflection images[12], and a modeled pressure source determined from geodetic data by removing fault-induced deformation for the 2015 eruption (Fig. 2c)[52]. These observations provide new details for previous findings: (1) the Cobb hotspot significantly enhances magma flux in the focused melt delivery pipe[5]; (2) the MMR is episodically replenished from deep sub-axial magma lens via porous flow and with varying magma supply rates during different eruption cycles[13]; and (3) although magma transport from the deep conduit to the shallow MMR provides pressure needed for triggering eruptions, our models suggest that the high-melt top layer of the MMR is the direct magmatic source (Figs. 2–5). A recent teleseismic study indicates that strong seismic attenuation below the Axial segment of the JdFR requires ~2% in situ melt in a < 50-km-wide and ~150-km-deep subridge region[54], which may serve as the melt source in the upper mantle for crustal focused magma delivery and accumulation beneath Axial Seamount.

## Origin of the asymmetric magma plumbing system beneath Axial Seamount

High-resolution bathymetry shows that most of the lava flows erupted in the last 800 years were extruded from fissures along the eastern edge of the caldera[8,14]. This is consistent with the eastward offset of the underlying MMR relative to the caldera and the northwestward dip of the magma chamber roof (Fig. 1b)[9]. The JdFR migrates westward at ~2 cm/a relative to the stationary and persistent Cobb hotspot plume[55]; the offset between the MMR and the summit caldera at Axial Seamount reveals continuous westward migration of the spreading center since the caldera formed ~1.1 kyr ago[56]. Due to ridge migration and trapdoor collapse of the caldera[5], the southeastern edge of the summit caldera is near the center of the 11-km crustal root associated with the Cobb hotspot and is located above the shallowest part of the MMR (Fig. 1b). Repeated magma transport to the seafloor helps form the diking pathway beneath the eastern caldera wall, where crustal accretion is mainly through lava eruptions from fissures and dike intrusions in the south and north rift zones. We infer that the WMR is a remnant melt-storage pocket left behind as the MMR moved eastward relative to the overlying caldera, and cooling and crystallization of magma in this pocket contributes to forming crust beneath the west caldera wall. This also agrees with the seafloor morphology of Axial Seamount, where the western shoulder is much wider, shallower, and thicker than the eastern one (Figs. 1b and 5).

One limitation of this study is that it uses one 2D seismic line across the summit of Axial Seamount. Additional insights might be gained by applying the analysis used here to other 2D lines of the 2002 MCS survey. This would test some of our ideas about how magma and partial melt is distributed in the shallow crust and the connections between them. For example, our FWI models suggest that the SMR is not directly connected to the MMR through shallow high-melt channels along the image of the J48 survey line. Rather, a deeper low-velocity anomaly appears to connect the SMR to the deep melt-mush feeder conduit. But we cannot rule out intermittent or permanent connection zones between the MMR and SMR that may be located outside of the image plane. In addition, observations of weaker seismic attenuation and moderate partial melt fraction in the SMR suggest that, unlike the MMR, its magma replenishment is more limited, and it is unlikely to trigger an eruption or intrusion event in the near future. This is consistent with the fact that few earthquakes were recorded around the SMR during recent eruption cycles[20,57]. With the continued migration of Axial Seamount relative to the underlying Cobb hotspot, the SMR is likely to experience enhanced magma supply in the future. But more 3D seismic surveys are needed over the southeast region of Axial Seamount to investigate 3D structure of the SMR and its connection to the MMR and deep magma supply conduits.

## Methods

### Data processing

The MCS field data was acquired during a 26-day EW0207 cruise in 2002 with the R/V Maurice Ewing to study the axial structure and crustal evolution of the JdFR[58,59]. The air-gun source, consisting of 10 elements with a total volume of 3005 cubic inches, was towed at a depth of ~7.5 m and fired every 37.5 m along the track. Each shot was recorded by 480 hydrophones with a 12.5-m interval at the depth of 10 m, and the minimum source-receiver distance is 190 m. A total of 10,265 shots were fired above Axial Seamount, and the data was sampled at 250 Hz for a record duration of 10.24 s. In this study, we used data from the J48 survey line crossing the Axial Seamount from the southwest to the northeast (Fig. 1). Prestack data processing includes bad trace killing and interpolation, random noise removal, bandpass filtering of 3–20 Hz, and 3D to 2D amplitude correction (3D field data records but 2D wave equation simulations and inversion are used in this study)[60]. A predictive deconvolution filter was used to suppress the bubble effects of the air gun sources. To reduce the mask effects of

deep water for refraction arrivals, a true-amplitude redatuming process is applied to the MCS streamer data to simulate on-bottom seismic experiments[5,26]; the receivers and sources were sequentially downward continued to the seafloor in common-source and common-receiver gathers[61]. The refractions in the redatuming data appear as the first arrivals, making it easy for fitting in traveltime tomography and FWI (Fig. S2).

### Viscoacoustic FWI

FWI is an advanced seismic imaging technology for high-resolution mapping subsurface model parameters, which directly fits observed data with synthetic records computed by solving the wave equation[62,63]. In exploration seismology, it has been successfully applied to characterize hydrocarbon reservoirs, with a resolution of tens to hundreds of meters[64,65]. In regional and global seismology, by fitting earthquake recordings[28,30,66] or ambient-noise crosscorrelation data[24,67,68], FWI has been used to image heterogeneous geological structures within the crust and mantle, with a resolution of a few to hundreds of kilometers. To account for seismic attenuation, we apply a viscoacoustic FWI scheme in this study to constrain both P-wave velocity and attenuation below the Axial Seamount[32]. The misfit function can be defined as

$$J(\mathbf{m}) = \| \mathbf{d}_{obs} - \mathbf{d}_{syn}(\mathbf{m})\|^2, \qquad (1)$$

where $\mathbf{m} = [V_p, Q_p]^T$, and $V_p$ and $Q_p$ denote the P-wave velocity and quality factor models, $\mathbf{d}_{obs}$ represents the redatuming MCS data, and $\mathbf{d}_{syn}(V_p, Q_p)$ denotes the synthetic data with sources and receives deployed on the seafloor. The synthetic data $\mathbf{d}_{syn}(V_p, Q_p)$ is computed by solving a time-domain complex-valued viscoacoustic wave equation that explicitly includes $Q_p$[35]. The misfit gradients are derived based on the adjoint-state method, and a preconditioned limited-memory Broyden-Fletcher-Goldfarb-Shanno (L-BFGS) algorithm[69] is used to reduce interparameter crosstalk and update $V_p$ and $Q_p$ models as

$$\mathbf{m}^{k+1} = \mathbf{m}^k(1 + \alpha\delta\mathbf{m}), \qquad (2)$$

where $k$ and $k+1$ denote the current and next iterations, $\delta\mathbf{m}$ is the updating direction calculated based on the L-BFGS algorithm, and $\alpha$ is the step length computed using a parabolic search method. The detailed implementation for the multi-parameter inversion can be found in ref. 32. To avoid the cycle skipping issue in FWI, we apply a multi-scale strategy to gradually incorporate high-frequency data[70]; three frequency bands are used, i.e., 3–5 Hz, 3–7 Hz, and 3–9 Hz (Figs. S2–S9). Numerical experiments show that the velocity perturbations have first-order effects on seismic data, while the attenuation perturbations have second- and even third-order effects. To mitigate large differences in data sensitivity to velocity and attenuation, we first update both $V_p$ and $Q_p$ models at the first stage and then update $Q_p$ model alone at the second stage. The detailed iteration numbers and data reduction at these two stages are presented in Fig. S6. The source wavelet is estimated for each source using the frequency-domain deconvolution as

$$F(\omega) = \frac{D_{obs}(\omega)G^*(\omega)}{G(\omega)G^*(\omega) + \epsilon}, \qquad (3)$$

where $\omega$ denotes the angular frequency, $D_{obs}(\omega)$ is the frequency spectrum of observed data, $G(\omega)$ is the frequency spectrum of the Green's function computed by solving the viscoacoustic wave equation with a zero-phase broadband wavelet, and $\epsilon$ is a small value to avoid division by zero. A cosine taper is applied to the estimated wavelet to attenuate oscillations after 2.5 periods. After the inversion in the frequency band of 3–5 Hz, the $V_p$ signatures for MMR, SMR, and hydrothermal boundary layer above the MMR become clearer than

those in the initial tomographic model, and the resolution of shallow layer 2A is improved (Fig. S7a). In the $Q_p$ model, a shallow strong attenuation layer associated with hydrothermal circulation and deep low-$Q_p$ region related to the MMR and magma feeding conduit have been recovered but with a lower resolution than the $V_p$ model (Fig. S7b, c). Increasing the data frequency to 9 Hz, the resolutions of $V_p$ and $Q_p$ models have been significantly improved, and small-scale magma reservoir features can be clearly imaged (Figs. S8 and S9).

## Calculation of temperature structure

Both anharmonic and anelastic effects have been considered to estimate the subsurface temperature structure beneath Axial Seamount based on FWI velocity and attenuation models. The derivative of P-wave velocity with respect to temperature can be written as[46,71]

$$\frac{\partial \ln V_p(\mathbf{x})}{\partial T} = \frac{\partial \ln V_p}{\partial T}\bigg|_{anhar} - \frac{F(\kappa)}{\pi Q_p(\mathbf{x})}\frac{H^*}{RT^2(\mathbf{x})}, \quad (4)$$

where $\mathbf{x}$ denotes the subsurface location, $Q_p$ is FWI quality factor model, $R = 8.314\,\mathrm{JK^{-1}mol^{-1}}$ is the universal gas constant, $T$ is the temperature, $F(\kappa) = (\pi\kappa/2)\cot(\pi\kappa/2)$, and $\kappa$ is the power law exponent of the frequency-dependent $Q_p$. $H^*$ is the activation enthalpy and is set to $276\,\mathrm{kJ\,mol^{-1}}$ based on the creep activation energy in the crust[72].

We computed two temperature models: (1) only assuming the anharmonic effect alone and (2) considering both anharmonic and anelastic effects, which approximate the minimum and maximum contributions of temperature to velocity perturbations[46]. In the first scenario, the first term on the right-hand-side of Eq. (4), i.e., $\frac{\partial \ln V_p}{\partial T}\big|_{anhar}$, is set to $-8.1e^{-5}\,\mathrm{K^{-173}}$ and the second term is set to zero. The temperature model can be calculated as

$$T(\mathbf{x}) = T_0(\mathbf{x}) + \delta \ln V_p(\mathbf{x}) / \frac{\partial \ln V_p(\mathbf{x})}{\partial T}, \quad (5)$$

where $T_0$ is a reference off-axis temperature model[74], $\delta \ln V_p = (V_p - V_{p0})/V_{p0}$ is the relative velocity perturbation (Fig. 2c), $V_p$ is the FWI velocity model, and $V_{p0}$ is the background velocity model (Fig. S1c). In the second scenario, $F(\kappa)$ is set to one to estimate the maximum anelastic effects[46]. Because the second term on the right-hand-side of Eq. (4) is temperature dependent, calculating the anelastic associated temperature model is a nonlinear problem. Following ref. 46, we iteratively update the temperature profile until the difference between adjacent iterations is less than 10 °C. The iteration scheme can be expressed as

$$T^{k+1}(\mathbf{x}) = T_0(\mathbf{x}) + \delta \ln V_p(\mathbf{x}) / \left[ \frac{\partial \ln V_p}{\partial T}\bigg|_{anhar} - \frac{1}{\pi Q_p(\mathbf{x})}\frac{H^*}{R\left[T^k(\mathbf{x})\right]^2} \right], \quad (6)$$

where $k$ and $k+1$ denote the current and next iterations. Estimated temperature models are shown in Fig. 4a and Fig. S15. Compared with the anharmonic model, high-temperature anomalies in the anelastic model are more focused around the deep melt conduit and magma reservoirs.

## Melt estimation

The thermal effect is not enough to explain the observed low velocity anomalies with temperature above 1150 °C. Thus, it is necessary to take partial melting into account. As shown in ref. 5, we apply a two-phase differential effective medium method to compute the melt fraction. The background matrix is assumed as solid basalt with P-wave velocity of 6.4 km/s, S-wave velocity of 3.5 km/s, and density of 2.97 g/cm³. The fluid phase is assumed as basaltic melt with P-wave velocity of 2.35 km/s, S-wave velocity of 0 km/s, and density of 2.7 g/cm³. Spherical and vertically aligned elliptical inclusions with aspect ratios of 1, 0.5, 0.1,

and 0.002 are used to simulate the effect of partial melt on P-wave velocity (Fig. 3). Based on this relation, we calculate subsurface melt fraction within the 1150 °C isotherm. For the case only considering the anharmonic effect, FWI velocity is directly used to predict the melt ratio. If incorporating the anelastic effect, the P-wave velocity is corrected according to the phase dispersion as

$$V_p(\mathbf{x},\omega) = V_p^{ref}(\mathbf{x})\left[1 + \frac{1}{\pi Q_p(\mathbf{x})}\ln\frac{\omega}{\omega_0}\right] \quad (7)$$

where $\omega_0 = 2\pi f_0, f_0$ denotes the reference frequency and is set to 1 Hz in viscoacoustic FWI[32], and $V_p^{ref}$ is the FWI velocity estimated at the reference frequency. The melt fraction models with different inclusion geometries are shown in Fig. 4a and Figs. 15–18. Spherical inclusions generate the largest melt fraction, as high as 37% in the upper layer of the MMR. The presence of elliptical inclusions significantly impacts estimated melt fractions, and the lower aspect ratio typically leads to smaller melt fraction. In addition to melt inclusion geometry, there are many other factors influencing the accuracy of estimated partial melt fractions. For instance, representing magma as an effective medium comprising solid basalt and basaltic melt is merely a two-phase simplification, whereas real magma might contain volatiles. Achieving a more realistic rock-physics representation for magma and an accurate relationship between seismic velocity and partial melt requires further investigation.

## Porosity estimation in the uppermost crust

Because of faults, fissures, pillow fragments, lava flows, and hydrothermal alteration, the uppermost oceanic crust has strong seismic attenuation[75,76]. Regression analysis for in situ log data from the ODP Hole 504B in the eastern equatorial Pacific showed that the crustal attenuation dominated by scattering effects from heterogeneities satisfies an exponential relation with porosity as[39]

$$Q^{-1} = Q_0^{-1}\exp(\beta\phi) \quad (8)$$

where $Q_0$ is a reference quality factor of 250 close to the intrinsic attention of deep crustal rocks[77], and $\beta = 25$ is the regression exponential coefficient. For low porosities, a linear model was also derived as[39]

$$Q^{-1} = Q_i^{-1} + Q_p^{-1}\phi \quad (9)$$

with $Q_i^{-1} = 0.0025$ and $Q_p^{-1} = 0.29$. Based on these two regression relations and the FWI $Q_p$ model, we compute the porosity profiles within seabed sediments and layer 2A (Fig. S13). The maximum porosity of ~10–16% is located in the areas of volcano flanks with large seafloor depth variations, and a slightly lower porosity of ~9–12% is present below the caldera. These porosity distributions are consistent with the shallow low-velocity perturbations, which might be caused by hydrothermal circulation in fractured basalt between the off-axis flank and axial caldera.

## Data availability

The multi-channel seismic data used in this study can be downloaded from the Marine Geoscience Data System (http://www.marine-geo.org/tools/search/entry.php?id=EW0207). The geological graph and imaging results are plotted using GMT (http://gmt.soest.hawaii.edu/projects/gmt/). The full waveform inversion models of the Axial Seamount and synthetic test datasets can be downloaded from https://github.com/JidongYangUPC/AxialSeamountFwiModel.

## Code availability

The code for viscoacoustic full waveform inversion can be accessed upon reasonable request from J. Yang (jidong.yang@upc.edu.cn).

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

## Acknowledgements

This research is supported by funding from the National Natural Science Foundation of China Outstanding Youth Science Fund Project (Overseas) (no. ZX20230152), the Natural Science Foundation of Shandong Province-General Program (ZR2023MD087), and the Marine S&T Fund of Shandong Province for Pilot National Laboratory for Marine Science and Technology (Qingdao) (no. 2021QNLM020001). Z.Z. is supported by National Natural Science Foundation of China projects of 2023002410 and 42374138. H.Z. is supported by the National Natural Science Foundation of EAR-2042098.

## Author contributions

J. Yang and H. Zhu designed the study and conducted the data processing and model inversion. Z. Zhao, J. Huang, D. Lumley, R. Stern, R. Dunn, A. Arnulf, and J. Ma contributed to the interpretation of the results, development of the concept model, and writing of the manuscript.

## Competing interests

The authors declare no competing interests.
