## [Peer Review File · Nature Communications]

Asymmetric magma plumbing system beneath Axial Seamount based on full-waveform inversion of seismic dataEditorial Note: Parts of this Peer Review File have been redacted as indicated to remove third-party material where no permission to publish could be obtained.

REVIEWER COMMENTS

Reviewer #1 (Remarks to the Author):

Review by William Chadwick

General comments on the manuscript:

This paper presents the results of re-processing one 2D multi-channel seismic (MCS) line collected at Axial Seamount in 2002 with a new method (full waveform inversion, FWI) that yields new 2D cross-sections of P-wave velocity and seismic attenuation with higher resolution than previously published. The paper then makes a series of new interpretations based on the new views along this one 2D cross-section. One of the main new interpretations is that there appear to be 4 distinct magma bodies below the summit of Axial Seamount, rather than the 2 previously imaged (which were one main magma reservoir below the summit caldera and a secondary one 5-10 km to the east below the SE flank of the volcano). The authors speculate on magma pathways between the 4 reservoirs, and between the main reservoir and the surface, based on narrow linear spatial variations of the P-wave velocity and attenuation in the 2D cross-sections. They also calculate melt-fraction in the various reservoirs and porosity in the crust.

One strength of the manuscript is that the authors thoroughly describe and reference relevant previous studies at Axial Seamount, and they attempt to explain and reconcile all recent data and observations with their proposed models and interpretations. The paper is well-written and the figures are good.

The main weakness of the manuscript is that the new results are only based on one single 2D seismic line from the 2002 survey (that included multiple lines), so it is unclear if what they observe extends into the 3rd dimension. Presumably the method they use could be applied to other 2D lines from the 2002 survey to see if the results are consistent with their present conclusions (and allowing the extrapolation of their results from one 2D line into 3D). The paper would be stronger and more convincing if the interpretations were supported by more than one 2D line. There is much more information that could be gained by considering the 3rd dimension (for example, spatial extent of the magma reservoirs, spatial variations melt fraction, etc)

Similarly, from the little that has been presented so far from a newer high-resolution 3D

MCS seismic survey at Axial Seamount in 2019 (for example, in a 2020 AGU abstract by Arnulf et al), the newer MCS results do not seem to support the conclusions in this paper (particularly the 2 “extra” magma reservoirs, nor the lateral connections between them). Can the FWI method be applied to the data from the 2019 survey? If so, what would it show? Would it support the conclusions presented here?

So this raises the fundamental question of how confident we can be that the 2D P-wave velocity and attenuation variations in this paper really show the distribution of partial melt in the 3D shallow crust at Axial? And what is the real resolution of these data in terms of discerning relatively small structures and melt conduits?

I think if the paper stays as the analysis of just one 2D MCS line, then it should treat the results as provisional, subject to confirmation by future analysis of other 2D lines from the 2002 survey, or analysis of data from the 2019 3D survey. Also, I think that some of the interpretations should be presented with less certainty and more as interesting possibilities than definite conclusions. For example, I think the existence and significance of the apparent “shallow LV zone” (=dike conduit?) is highly debatable and not very convincing. For this reason, I do not think it should be the focus of the Abstract (which should be completely re-written) or highlighted in the title. Instead, I would highlight the increased resolution of the seismic imaging method, the provisional interpretation of multiple shallow magma reservoirs compared to earlier interpretations, the possible imaging of connections between them, and the deeper conduit below them. But it also important that the paper clearly state that confirmation of these interpretations must await additional evidence from further studies.

I recommend a re-evaluation of the paper after major revisions to address the editorial suggestions and specific comments in the annotated manuscript, some of which are listed below.

Specific comments (keyed to lines in the annotated manuscript):

Line 13 (Abstract): To me, the Abstract does not do a good job of summarizing the key results. I would not make the eastern offset of recent eruptive fissures an emphasis in (so I would omit the 2nd sentence). I think a more important conclusion to highlight in the Abstract, is that you image more magma reservoirs in the shallow crust compared with

earlier analysis of this same data set. I think the Abstract should be re-written with that emphasis. In the 3rd sentence, you need to say you are only re-analyzing one 2D line from the 2002 MCS survey. The 4th sentence about melt fractions makes it sound like the "deeper feeding conduit" is part of the MMR, but in the rest of the paper it is treated as something separate. Rephrase to be consistent. I would re-write the last 3 sentences of the Abstract to focus on the magma reservoir results rather than the "LV throat" and the eastern offset of recent eruptions.

Line 21: I think your reference #28 (Carbotte et al., 2008) might be a more appropriate here than your #6 (Canales et al., 2009).

Line 25: You could add reference #56 here for the 2015 eruption.

Line 28: Reference #12 is not directly related to the details in this sentence. However, it could be used at the end of the previous sentence if you want.

Line 30: Again, not sure reference #6 (Canales et al., 2009) is relevant here.

Line 32: Not sure reference #16 (Dziak et al., 2011) is relevant here.

Line 36: The end of this sentence doesn't make sense because the eastern wall *is* seismically active.

Line 39: It's important to put this West et al. (2001) study in the right context. It was the first to use seismic tomography to attempt to image the magmatic system below Axial, so is significant. But its results now seem primitive and coarse compared to the much higher resolution methods used by Arnulf et al. (2014; 2018) from the 2D MCS survey in 2002 and the 3D MCS survey in 2019 (still unpublished except for AGU abstracts in 2019 and 2020).

Lines 42-44: Yes, but for both of these studies the melt fraction numbers are average values over a very large volume.

Line 47: You should reference where these numbers come from and clarify that they include estimates of the volume of lava flows and subsurface dikes.

Line 48: In Arnulf et al. (2014), which is reference #13, they interpreted a high melt zone at the top of the MMR beneath the southern part of the caldera, so a "high-melt zone" has been mapped before, but they did not assign any quantitative melt fractions (just "high" and "low", or "melt" and "mush"). Arnulf et al. (2018) placed some constraints on the melt fractions. So saying it's "...unclear whether there is a high-melt zone..." doesn't seem accurate. What are you trying to say here exactly?

Line 56: Whether it is easy or not is subjective. What are the authors really trying to say here? Should be rephrased. Is there an simple way to explain what "downward continued to the seafloor" means for readers who are not seismic experts?

Line 58: You should note that it was called the "SMR" in Arnulf et al. (2018), ref #9 here, and that you are calling it something different here ("EMR").

Line 64: In the caption for Figure S1, explain what Q_p is and how it relates to the figure. Also can you add a plain language explanation of what "viscoacoustic" means in this context?

Line 65: Cross-sections of "attenuation" are important in the rest of the paper. I think it would help to take a sentence or two here to explain what "attenuation" means in this context and what these cross-sections show.

Line 72: What data are fit better? Need to better explain this.

Line 75: How much of a velocity anomaly or how low a low-Q anomaly? And at what depths? Is this claim substantiated in the figures in the Supplementary materials? If so, which figures?

Lines 82-84: To me, this sentence is poorly constrained speculation. "Hydrothermal cooling above the MMR" is a pretty generic explanation and is unconvincing. I can think of other

interpretations (the existence of thick, dense, ponded lava flows in the subsiding caldera floor block, for example), but I don't think there is compelling evidence for any one idea. Likewise, the low-velocity boundaries could be just due to the caldera faults. There is nothing that convincingly identifies them as "magma pathways". In fact, I might expect a zone of diking to have higher velocities. Then there is the problem that zones of diking are likely vertical and narrow, so would be hard to image seismically. Also, the references cited in this sentence seem generic and do not particularly substantiate the interpretations. For all these reasons, I would omit this sentence.

Line 85: Does attenuation (Q_p) have units?

Lines 86-88: How does hydrothermal alteration "induce" high-porosity? Not obvious to me. Cracking itself & deformation make high porosity. Or the extrusive layer itself starts with high porosity.

Line 95 (Figure 2): In Fig. 2, I would change the y-axis labels to "depth below sealevel" to clarify that it is not depth below the seafloor. What is the "perturbed V_p model" in Figure 2c? And what is the difference between Fig. 2a and c? This needs some explanation if this is what you are referring to as what is showing "more detail". Is it just imaging the MMR in more detail? Or is it imaging multiple magma reservoirs in the shallow crust in more detail? Also, I think the blue contours in Figure 2c deserve some explanation in the text. What is interpreted that this contour signifies and what is it based on? I would also add a plain language explanation of what Fig. 2b shows and why it makes intuitive sense (if it does). What is "seismic attenuation" exactly, and why isn't there major (and more uniform) attenuation inside the MMR?

Line 96: You need to indicate which panel in Figure 2 the reader should refer to to see this. Is this the red/white "smile" at 2km depth in Fig. 2c??? Also is the "P-wave speed anomaly" fast or slow? And is it really "at the top" of the MMR, or above it?

Line 98: Wait a minute ... there is "high-melt fraction" in the pink "smile" at ~500 m depth below the caldera??? I thought the magma was in the MMR at ~1 km depth below the

seafloor. This needs clarification.

Line 103: What does this mean or imply and how does it make sense?

Line 108: I would say ~7 km east of the MMR, if you measure from the center of the MMR to the center of the EMR.

Lines 111-112: Depth below sealevel? Or depth below the seafloor?

Line 117: Can this method *really* identify something of this scale at this depth? How do you know this is actually a fluid conduit?

Line 118: I can't help but wonder how confident can we be of this interpretation... This is just one 2D seismic line. Would these "extra" low-V zones interpreted as magma reservoirs be imaged on other lines from the 2002 MCS survey? The evidence would be stronger if you saw the same things on other 2D lines and that made sense in three dimensions.

Lines 118-121: I would say that the distribution of seismicity on the caldera ring fault doesn't necessarily say anything about whether the WMR could feed a dike to the surface in the western part of the caldera. That said, there aren't many eruptive fissures there from the last several hundred years from the mapping of Clague et al. (2013). In general I think it's dubious that MCS methods can image narrow vertical structures like dikes in the shallow crust. I would not expect to see them, even if they do exist.

Lines 121-124: I'm equally dubious of this interpretation. I'm not convinced this "LV throat" is real or significant. The seismicity shows slip along the caldera faults during inflationary or deflationary deformation. It doesn't necessarily say anything about diking or the existence of a magma pathway there. In fact, one might expect a "diking pathway" to have a high velocity. I'm not convinced that MCS can image narrow vertical structures like "dike pathways".

Line 141: I wonder why there isn't a high melt fraction at the very top (roof) of the MMR in

Figure 4b. Isn't that where the MCS data show the highest seismic reflections indicative of a change from solid to melt?

Line 143: What you are calling the "upper MMR" in Figure 4, looks like the middle of the MMR to me, based on the pink and blue outlines.

Line 145: Is the volume of melt in the "upper MMR" a subset? If so, what percent of the MMR is the "upper MMR"? How can you constrain the volume of the "upper MMR", based on only one 2D MCS line? How do you know the 3rd dimension for calculating a volume? Needs clarification.

Line 147: Note reference #11 is duplicated as reference #43.

Line 150: It's still unclear what constitutes the "upper" vs. "lower" MMR.

Line 162: What is the "boundary layer" in Figure 5?

Line 168-170: I'm not convinced the data show this, especially the "LV throat". You could have drawn these features on Figure 5 before this study was done, because it was already obvious there is some sort of focusing of magma from the top of the MMR into dikes that intrude upward near the eastern caldera fault, but I'd say the reasons for that are still unclear. To me, the results in this paper do not illuminate why that is the case.

Lines 173-178: This explanation for the difference between the eastern and western sides of the caldera seems very speculative...

Line 216: Actually, Clague et al. now think the present caldera only formed about 1100 years ago (the age reported in Clague et al. (2013) is incorrect). This was presented in an AGU abstract:

Clague, D. A., R. A. Portner, J. B. Paduan, M. Le Saout, and B. M. Dreyer (2019), Formation of the summit caldera at Axial Seamount. Abstract presented at 2019 Fall Meeting, AGU, San

Francisco, CA, 9-13 Dec.

Lines 223-225: This is an interesting idea, but what would explain the EMR?

Lines 229-232: This is the first acknowledgement that all the results in this paper are based on only one 2D seismic line and that this could limit what you can conclude! In my opinion, there should be much more of this type of discussion of the limitation of this study, and what would be needed to better substantiate the tentative results presented.

Another important question that should be addressed in the paper is: if they are real, why were the WMR and ESMR (the 2 “extra” magma reservoirs) not previously identified by seismic reflection methods in the 2002 (and 2019) MCS surveys? To me, seismic reflection is a more compelling way to identify where magma bodies exist in the crust, because of the strong reflections produced by the change from solid rock to liquid, whereas mapping seismic velocities seems like a less definitive and more indirect way of doing the same thing. Why should readers believe these new results based on only seismic velocities if they are not confirmed by reflection seismology?

Reviewer #2 (Remarks to the Author):

This is an important and excellent paper and may be accepted with minor revision.

Important contributions of the paper include P-velocity imaging with much higher spatial resolution than previous work and novel synoptic attenuation imaging. Taken together, these advances allow the authors to address connectivity (or lack thereof) between shallow magma systems and to obtain more accurate estimates of the melt fraction. The former is important to understanding how the shallow magma plumbing influence eruptive style (eruptions being the arguably the best monitored of any submarine volcano). The latter is important because of the perceived discrepancy between previous estimated of melt fraction and that thought, on experimental grounds, needed to promote robust magma storage.

My suggestion for a minor revision concerns the discussion of melt fraction. I confess that I have a hard time wrapping my mind about quantifications like “ranging from 39% to 65%” (line 144).

First, melt fraction is likely a matter of spatial scale. I presume that at the finest scale, the maximum is 100%, meaning that somewhere in the magma system is a little pocket of pure melt. So, a number like 39% is only meaningful in the sense of representing an average with respect to some volume of material (rock+melt). As the imaging is tomographic, I suppose that the relevant averaging volume ought to be based on the volume of the resolving kernel. The Supplementary Material mentions resolution in connection with Figs 10-12, but I was unable to discern how to use them to figure out this volume. After pondering the issue for a while, I would like to make the suggestion that the authors add an “exceedance plot” like this:

Here, X is informed by some estimate of the resolving power of the tomography. Such a cumulative histogram would provide more information than what's in line 144. It could be generated very easily from the model underpinning the author's Fig 4b.

I also confess that ranges like “39% to 65” (line 144) raise more questions with me than they answer. Is the range estimate a confidence interval, e.g., 52% +/- 13? If so, what is the formal confidence? Is it 95%, or something else? What sources of variation contribute to this confidence, and has “model error” been included. I imagine that model error (including the ability to map Vp and Qp into melt fraction) is the main source of error. But speaking only for myself, and not for the authors, I have no confidence in my ability to quantify such an error. Anyway, I think that the authors ought to explain what they intend a range to mean. Such an explanation could be based on the exceedance plot (as in my figure). I also wonder whether an intercomparison of a range with that stated in another paper (as the authors provide, e.g., in line 40) is really as straightforward as it seems. This point could be addressed with a few additional lines of text.

Reviewer #3 (Remarks to the Author):

This manuscript presents a 2-D full waveform inversion for VP and QP along a profile that crosses the summit caldera in WSW-ENE orientation. The inversion is technically impressive and reveals some interesting features, specifically a low velocity pathway in the location where dikes are emplaced along the eastern side of the caldera to feed eruptions from the main magma chamber, a western magma reservoir that has not been previously reported, a shallow high velocity region infilling the caldera and a narrow region of low velocities underlying the main magma chamber. The presentation of the results is somewhat jumbled, and the interpretation does an inadequate job of identifying what has been learned that is new and what is just confirming previous ideas. In discussing the connection between magma bodies, the authors seem to be inferring 3-D geometry from a 2-D inversion – a 2-D inversion can show bodies are connected but not demonstrate that they are not. It is impressive that the inversions can image the magma pathway on the eastern wall, but we know from previous studies that is where eruptions occur because that is where the plate boundary is, so what is new in the interpretation? The paper does develop some ideas about the asymmetry of the magma chamber relative to the caldera and its influence on the location of eruptions. There is also speculation that the connectivity to the western magma body may play a role in supporting asymmetric on the caldera fault and several short deflation events observed from 2016 to 2019, but the manuscript does not make any attempt to test this idea with the geodetic data. With some thought and more reference to the prior literature, I think this manuscript could be revised into an inciteful publication.

Detailed comments:

14-51. It is not clear what questions are motivating this study?

34-36 “Although previous studies revealed the location of the subcaldera magma chamber, it remains unclear why all recent eruptions were only from the eastern wall rather than the two seismically active walls” – I disagree, it is very clear why recent eruptions are on the east wall. This is where the plate boundary is and where the summit connects to the north and south rifts, one of which also spreads in each eruption and takes most of the melt volume. Given the location of the plate boundary (e.g., Chadwick et al., 2013), how can a

spreading event be supported by an eruption on the west wall? It is interesting that the asymmetric offset of the main magma chamber relative to the caldera which was first reported by Arnulf et al. (2014) centers it beneath the plate boundary.

48-50. "it is unclear whether there is a high-melt zone beneath the caldera with a melt fraction close to the eruption threshold and a volume comparable to the size of recent magmatic events" – If it is unclear what is the alternative? This is the only mechanism that I am aware of to explain eruptions on mid-ocean ridges and the presence of a shallow high-melt zone supporting eruptions has been demonstrated along the East Pacific Rise (e.g., Singh et al., 1998; Xu et al., 2022) and the Juan de Fuca Ridge (Canales et al., 2006)

79-80. What is the significance of the shallow high velocity body in the caldera? At least in some other calderas, the caldera infill has a very low velocity (e.g., Hooft et al., 2019;). I think a review of prior observations in calderas would be useful to interpret this result which is presently not interpreted

80. "246 m" - To this level of precision?

81-82. The inward dipping normal faults are mainly identified by the caldera walls. Most of the seismicity is on the buried outward dipping ring fault. While both Arnulf et al. (2018) and Waldhauser et al. (2020) associate some earthquakes with the inward dipping fault on the eastern side of the caldera, the catalogs are inconsistent.

82-84. The first half of this sentence seems to be referring to structure right above the magma body which doesn't seem to be mentioned anywhere else. The second half seem to be referring to the feature that is presented in more detail in line 121-124. The whole sentence is orphaned in a section on shallow structure and thus quite misplaced.

110-114. Since the center of the EMR is well to the south of the MMR (Arnulf et al., 2014) and this profile, there is no justification for concluding it lacks connectivity to the MMR from a 2-D model. I think this is a major flaw in the interpretation.

117. Is the WMR seen in the reflection data? If not, why not given the sharp velocity gradients near its top?

119-121. I do not follow. Geometrically, the fault must terminate in a mushy zone that can deform viscously so this could be a magma conduit. From the plots in Figure 4b and S15-18, it looks suspiciously like the earthquakes do root in the combined melt body producing a notch in it, potentially because of enhanced cooling.

121. "In addition". Should this be "Second" based on lines 114-115.

122-124. How does this pathway compare with those inferred by Arnulf et al. (2014) from a 3-D inversion?

145. How do the authors get this estimate? In Figures 4b/S15 the 32% region looks to be 0.2 km thick and 3 km wide so for the upper limit of 0.57 km³, this requires a 3rd dimension of $0.57 / (3 \times 0.2 \times 0.32) = 3$ km but the magma chamber extends 15 km along axis. If the 26% contour was used, there would be much more melt, so why select the 32% contour other than to get the answer that is desired? Some clarification in supplementary docs would be helpful.

156-159. How does this interpretation relate to previous interpretations of magma chambers on mid-ocean ridges? Isn't this the long-standing interpretation of crustal magma chambers at mid ocean ridges (Sinton and Detrick, 1992; Singh et al., 1998)?

173-176. If magma transport in and out of the WMR relaxes stresses why are there remaining stresses? Hefner et al. (2020) have modeled caldera inflation with asymmetric fault motions and show the remaining caldera inflation is quite symmetric, so I am not sure what need to be explained by transport of magma in and out of the WMR? The fault motions are asymmetric because the center of inflating magma body is offset to the east of the caldera.

178-180. This is one reasonable explanation and that adopted by Hefner et al. (2020), but an

alternative explanation is that the west wall is less coupled.

180.-182. The short-term deflation events from 2016-2019 are intriguing and loss of magma to the WMR is a feasible explanation, but such short-term deflation events were not seen between the 1998 and 2011 eruptions and the 2011 and 2015 eruptions so they are not related to a process that is inherent to the eruptive cycle as the model presented here seems to suggest. If these events are related to loss of magma into the WMR, then they should be geodetically observable by comparing the pressure gauge and tilt meter from the cabled instruments at the ASHES vent field on the western side of the caldera with those elsewhere. Have the authors looked?

186-188. The lack of a magma pathway is consistent with magma not erupting along the western wall, but it is not an explanation of why there is no pathway.

189-191. This is an odd choice of citations and prior work does not predict a broad magma body – indeed quite the opposite. At the EPR Dunn et al. (2000) report a width that “is narrow in the {lower} crust (5–7 km)” at 2-6 km depth and given the limits of travel time tomographic resolution, it presumably could be even narrower. Indeed, in a more recent paper Dunn (2022) estimates a width of 4.5 km at shallower depths broadening somewhat in the lower crust. Arnoux et al. (2019) also conclude the magma chamber is narrow in the lower crust along the Endeavour segment of the Juan de Fuca Ridge.

The model in this manuscript only extends to 2.5 km bsf so it not at all clear how wide the low velocity zone is at greater depth. The inversions in this manuscript have higher resolution and thus resolve structure beneath the MMR that not been resolved previously but it is not clear that if the image extended to greater depth, it wouldn't be anything other than very consistent with the prior travel time tomography studies.

201-204. Where else other than the upper mantle can the melt come from? What is purpose of this sentence.

221-227. This is interesting.

228-229. As noted above a 2-D image cannot show this.

William Wilcock

Arnulf, A. F., Harding, A. J., Kent, G. M., Carbotte, S. M., Canales, J. P., & Nedimović, M. R. (2014). Anatomy of an active submarine volcano. *Geology*, 42(8), 655-658.

Arnulf, A. F., Harding, A. J., Kent, G. M., & Wilcock, W. S. D. (2018). Structure, seismicity, and accretionary processes at the hot spot-influenced Axial Seamount on the Juan de Fuca Ridge. *Journal of Geophysical Research: Solid Earth*, 123(6), 4618-4646.

Arnoux, G. M., Toomey, D. R., Hooft, E. E. E., & Wilcock, W. S. D. (2019). Seismic imaging and physical properties of the Endeavour segment: Evidence that skew between mantle and crustal magmatic systems governs spreading center processes. *Geochemistry, Geophysics, Geosystems*, 20(3), 1319-1339.

Canales, J. P., Singh, S. C., Detrick, R. S., Carbotte, S. M., Harding, A., Kent, G. M., et al. (2006). Seismic evidence for variations in axial magma chamber properties along the southern Juan de Fuca Ridge. *Earth Planet. Sci. Lett.* 246 (3-4), 353–366.
doi:10.1016/j.epsl.2006.04.032

Chadwick Jr, W. W., Clague, D. A., Embley, R. W., Perfit, M. R., Butterfield, D. A., Caress, D. W., ... & Bobbitt, A. M. (2013). The 1998 eruption of Axial Seamount: New insights on submarine lava flow emplacement from high-resolution mapping. *Geochemistry, Geophysics, Geosystems*, 14(10), 3939-3968.

Chadwick Jr, W. W., Wilcock, W. S., Nooner, S. L., Beeson, J. W., Sawyer, A. M., & Lau, T. K. (2022). Geodetic monitoring at Axial Seamount since its 2015 eruption reveals a waning magma supply and tightly linked rates of deformation and seismicity. *Geochemistry, Geophysics, Geosystems*, 23(1), e2021GC010153.

Dunn, R. A., Toomey, D. R., & Solomon, S. C. (2000). Three-dimensional seismic structure and physical properties of the crust and shallow mantle beneath the East Pacific Rise at 9° 30'N. *Journal of Geophysical Research: Solid Earth*, 105(B10), 23537-23555.

Dunn, Robert A. "A Dual-Level Magmatic System Beneath the East Pacific Rise, 9° N." *Geophysical Research Letters* 49.18 (2022): e2022GL097732.

Hooft, E. E. E., Heath, B. A., Toomey, D. R., Paulatto, M., Papazachos, C. B., Nomikou, P., ... & Warner, M. R. (2019). Seismic imaging of Santorini: Subsurface constraints on caldera collapse and present-day magma recharge. *Earth and Planetary Science Letters*, 514, 48-61.

Hussenoeder, S. A., Collins, J. A., Kent, G. M., and Detrick, R. S. (1996). Seismic analysis of the axial magma chamber reflector along the southern East Pacific Rise from conventional reflection profiling. *J. Geophys. Res.: Solid Earth* 101 (B10), 22087–22105.

Singh, S. C., Kent, G. M., Collier, J. S., Harding, A. J., & Orcutt, J. A. (1998). Melt to mush variations in crustal magma properties along the ridge crest at the southern East Pacific Rise. *Nature*, 394(6696), 874-878.

Sinton, J. M., & Detrick, R. S. (1992). Mid-ocean ridge magma chambers. *Journal of Geophysical Research: Solid Earth*, 97(B1), 197-216.

Waldhauser, F., Wilcock, W. S. D., Tolstoy, M., Baillard, C., Tan, Y. J., & Schaff, D. P. (2020). Precision seismic monitoring and analysis at Axial Seamount using a real-time double-difference system. *Journal of Geophysical Research: Solid Earth*, 125(5), e2019JB018796.

Xu, M., Pablo Canales, J., Carbotte, S. M., Carton, H., Nedimović, M. R., and Mutter, J. C. (2014). Variations in axial magma lens properties along the East Pacific Rise (9° 30' N–10° 00' N) from swath 3-D seismic imaging and 1-D waveform inversion. *J. Geophys. Res. Solid Earth* 119 (4), 2721–2744. doi:10.1002/2013jb010730

Dear reviewers:

This is our response to the reviews of manuscript, entitled “Asymmetric magma plumbing beneath Axial Seamount, Juan de Fuca Ridge”. Comments from editors and reviewers are in red italics and our response is in black normal type. Page numbers of the modification corresponded to the comments are at the beginning of each response. A revised version with tracked changes is uploaded for supporting review, in which each response is marked at the margin. All figure numbers refer to the resubmitted manuscript, unless noted otherwise. Bibliographic references can be found in the manuscript.

I. Responses for Reviewer-1

1.1: The paper is well-written and the figures are good. The main weakness of the manuscript is that the new results are only based on one single 2D seismic line from the 2002 survey (that included multiple lines), so it is unclear if what they observe extends into the 3rd dimension. Presumably the method they use could be applied to other 2D lines from the 2002 survey to see if the results are consistent with their present conclusions (and allowing the extrapolation of their results from one 2D line into 3D). The paper would be stronger and more convincing if the interpretations were supported by more than one 2D line. There is much more information that could be gained by considering the 3rd dimension (for example, spatial extent of the magma reservoirs, spatial variations melt fraction, etc).

Response: We agree with the reviewer that 3D imaging for Axial Seamount can give a more comprehensive view than 2D seismic analysis. Limited by the available data and the expensive computational requirements of 3D waveform inversion, 2D seismic velocity inversion and reflection imaging are commonly used for studying the structure of mid-ocean ridges. Representative publications include:

- (1) Wilcock, W.S., Solomon, S.C., Purdy, G.M. and Toomey, D.R., 1992. The seismic attenuation structure of a fast-spreading mid-ocean ridge. *Science*, 258(5087), pp.1470-1474.
- (2) Team, T.M.S., 1998. Imaging the deep seismic structure beneath a mid-ocean ridge: The MELT experiment. *Science*, 280(5367), pp.1215-1218.
- (3) Guo, P., Singh, S.C., Vaddineni, V.A., Grevemeyer, I. and Saygin, E., 2022. Lower oceanic crust formed by in situ melt crystallization revealed by seismic layering. *Nature geoscience*, 15(7), pp.591-596.
- (3) Vaddineni, V.A., Singh, S.C., Grevemeyer, I., Audhkhasi, P. and Papenberg, C., 2021. Evolution of the crustal and upper mantle seismic structure from 027 Ma in the equatorial Atlantic Ocean at 2 43 S. *Journal of Geophysical Research: Solid Earth*, 126(6), p.e2020JB021390.
- (4) Holmes, R.C., Tolstoy, M., Cochran, J.R. and Floyd, J.S., 2008. Crustal thickness variations along the Southeast Indian Ridge (100116 E) from 2D body wave tomography. *Geochemistry, Geophysics, Geosystems*, 9(12).
- (5) White, D.J. and Clowes, R.M., 1990. Shallow crustal structure beneath the Juan de Fuca Ridge from 2-D seismic refraction tomography. *Geophysical Journal International*, 100(3), pp.349-367.
- (6) Dannowski, A., Grevemeyer, I., Ranero, C.R., Ceuleneer, G., Maia, M., Morgan, J.P. and Gente, P., 2010. Seismic structure of an oceanic core complex at the MidAtlantic Ridge, 22 19 N. *Journal of Geophysical Research: Solid Earth*, 115(B7).

According to the reviewer's suggestion, we have incorporated more analysis into the updated manuscript. The 3D geometries for the main magma reservoir (MMR) and the high-melt upper MMR are shown in Fig. S21, extracted from the tomography model proposed by Arnulf et al. (2018). The description regarding the connection between the MMR and other magma reservoirs may be inaccurate in the original manuscript. We have rewritten and revised them in the updated version. Furthermore, to validate our models, we have carefully analyzed the reflection images of Arnulf et al. (2014) and Carbotte et al. (2020), the geodetic data in Chadwick et al. (2022), the 3D tomography velocity model by Arnulf et al. (2018), and slip along outward-dipping fault planes from Hefner et al. (2020). Detailed discussions can be found in the responses to questions 1.33, 1.47, 3.2, 3.11, 3.14, 3.18, and 3.19.

1.2: Similarly, from the little that has been presented so far from a newer high-resolution 3D MCS seismic survey at Axial Seamount in 2019 (for example, in a 2020 AGU abstract by Arnulf et al), the newer MCS results do not seem to support the conclusions in this paper (particularly the 2 "extra" magma reservoirs, nor the lateral connections between them). Can the FWI method be applied to the data from the 2019 survey? If so, what would it show? Would it support the conclusions presented here?

Response: At the current stage, we do not have access to the 3D MCS seismic dataset. Upon reviewing the reflection images in Carbotte et al. (2020), we observed some coherent reflection events at the locations of the newly identified magma reservoirs in the shallow crust. But previous studies did not interpret these reflections in detail. In addition, we examined the tomography velocity model and geodetic data from published papers (Arnulf et al., 2018, Chadwick et al., 2022), which support our proposed model with multiple magma reservoirs in the shallow crust. Regarding the 2019 3D dataset, we will discuss the data sharing process with Dr. Adrien Arnulf. If we gain access to this data in the future, we will try our best to conduct high-resolution 3D FWI to image the detailed velocity and attenuation structures beneath Axial Seamount.

1.3: *So this raises the fundamental question of how confident we can be that the 2D P-wave velocity and attenuation variations in this paper really show the distribution of partial melt in the 3D shallow crust at Axial? And what is the real resolution of these data in terms of discerning relatively small structures and melt conduits?*

Response: The Rayleigh criterion, which states that the resolution of seismic imaging is approximately one fourth of the seismic wavelength, is commonly used as a measure of resolution. Based on the velocity distribution, the resolution is approximately 83 m in the shallow layer with a V_p of 3 km/s and around 167 m in a deeper layer with a V_p of 6 km/s along the 2D plane. But because of using 2D inversion, the uncertainty out of the plane is relatively large, with a resolution limit up to ten wavelengths (~ 3 -6 km). Regarding the reliability of 2D inversion results, we conducted tests using different parameters and optimization methods. The presence of multiple reservoirs in the shallow crust and a low-velocity throat are stable features. Therefore, the results along the J48 survey line are considered reliable. In seismology and mid-ocean ridge studies as shown in the response to question 1.1, 2D imaging is commonly employed and the results can provide important insights into subsurface structure and geological information. Based on our high-resolution 2D models and previous 3D tomography model, we can extrapolate the 2D structures to 3D to give a general view of melt distribution.

1.4: *I think that some of the interpretations should be presented with less certainty and more as interesting possibilities than definite conclusions. For example, I think the existence and significance of the apparent “shallow LV zone” (dike conduit?) is highly debatable and not very convincing. For this reason, I do not think it should be the focus of the Abstract (which should be completely re-written) or highlighted in the title.*

Response:[Abstract] On reflection images, the nearly vertical dike conduits are very difficult to resolve, because useful signals cannot be recorded at surface (see the black line in the following figure). However, FWI is a high-resolution quantitative inversion method for estimating the subsurface physical property (velocity), which utilizes all kind of wave types, including refractions (yellow line), diving waves (red line) and scatterings (purple lines), thus having the potential to constrain vertical and fine-scale structures. In seismology, there are many successful applications of FWI to map the mudstone and salt diapirs (Xiao et al, 2016, Interpretation, An offshore Gabon full-waveform inversion case study; Wang et al, 2019, The Leading Edge, Full-waveform inversion for salt: A coming of age). According to the reviewer’s suggestion, we have rewritten the abstract, without highlighting the shallow low-velocity zone.

1.5: Instead, I would highlight the increased resolution of the seismic imaging method, the provisional interpretation of multiple shallow magma reservoirs compared to earlier interpretations, the possible imaging of connections between them, and the deeper conduit below them. But it also important that the paper clearly state that confirmation of these interpretations must await additional evidence from further studies.

Response:[Abstract, p. 6, 11] According to the reviewer’s suggestion, we have rewritten the abstract and emphasize the multiple magma reservoirs in the shallow crust in the update manuscript. We also added more descriptions about the necessity to do more 3D investigation in the future.

1.6: Line 13 (Abstract): To me, the Abstract does not do a good job of summarizing the key results. I would not make the eastern offset of recent eruptive fissures an emphasis in (so I would omit the 2nd sentence). I think a more important conclusion to highlight in the Abstract, is that you image more magma reservoirs in the shallow crust compared with earlier analysis of this same data set. I think the Abstract should be re-written with that emphasis. In the 3rd sentence, you need to say you are only re-analyzing one 2D line from the 2002 MCS survey. The 4th sentence about melt fractions makes it sound like the “deeper feeding conduit” is part of the MMR, but in the rest of the paper it is treated as something separate. Rephrase to be consistent. I would re-write the last 3 sentences of the Abstract to focus on the magma reservoir results rather than the “LV throat” and the eastern offset of recent eruptions.

Response:[Abstract] Following this suggestion, we rewrote the abstract in the updated manuscript, removing the second sentence and emphasizing the magma reservoir results.

1.7: Line 21: I think your reference 28 (Carbotte et al., 2008) might be a more appropriate here than your reference 6 (Canales et al., 2009).

Response:[p. 2] We have changed the reference of Canales et al. (2009) to a more appropriate reference of Carbotte et al. (2008) in the revised manuscript.

1.8: Line 25: You could add reference 56 here for the 2015 eruption.

Response:[p. 2] We have added the reference of Chadwick et al. (2016) here in the updated manuscript.

1.9: Line 28: Reference 12 is not directly related to the details in this sentence. However, it could be used at the end of the previous sentence if you want.

Response:[p. 2] We have adjusted the references of West et al. (2001), Arnulf et al. (2014) and Arnulf et al. (2018) in these two sentences.

1.10: Line 30: Again, not sure reference #6 (Canales et al., 2009) is relevant here.

Response:[p. 2] We have removed the reference of Canales et al. (2009) here in the revised manuscript.

1.11: Line 32: Not sure reference #16 (Dziak et al., 2011) is relevant here.

Response:[p. 2] We have removed the reference of Dziak et al. (2012) here in the revised manuscript.

1.12: Line 36: The end of this sentence doesn't make sense because the eastern wall is seismically active.

Response:[p. 2] We have revised this sentence based on the reviewer's suggestion to make it clear and reasonable.

1.13: Line 39: It's important to put this West et al. (2001) study in the right context. It was the first to use seismic tomography to attempt to image the magmatic system below Axial, so is significant. But its results now seem primitive and coarse compared to the much higher resolution methods used by Arnulf et al. (2014; 2018) from the 2D MCS survey in 2002 and the 3D MCS survey in 2019 (still unpublished except for AGU abstracts in 2019 and 2020).

Response:[p. 2] We have rewritten this sentence according to the reviewer's annotation and adjusted the location in the text to describe the significance and limitation of West et al. (2001)'s work.

1.14: Lines 42-44: Yes, but for both of these studies the melt fraction numbers are average values over a very large volume.

Response:[p. 3] We agree with the reviewer about this point. The melt fractions estimated in these two studies are average values over a large volume, which might have large uncertainty. We have revised this sentence to make it more reasonable in the updated manuscript.

1.15: Line 47: You should reference where these numbers come from and clarify that they include estimates of the volume of lava flows and subsurface dikes.

Response:[p. 3] We have clarified the magma volume of seafloor lava flow and subsurface dikes, and cited related references in the updated manuscript.

1.16: Line 48: In Arnulf et al. (2014), which is reference #13, they interpreted a high melt zone at the top of the MMR beneath the southern part of the caldera, so a “high-melt zone” has been mapped before, but they did not assign any quantitative melt fractions (just “high” and “low”, or “melt” and “mush”). Arnulf et al. (2018) placed some constraints on the melt fractions. So saying it’s “...unclear whether there is a high-melt zone ...” doesn’t seem accurate. What are you trying to say here exactly?

Response:[p. 3] We carefully checked the papers of Arnulf et al. (2014) and Arnulf et al. (2018) and agree with the reviewer’s statement. Dr. Adrien Arnulf is also our co-author. The description in the original manuscript might be not accurate. Here we would like to say that previous studies did not provide accurate and quantitative estimations of the melt fractions and magma volume in the high-melt zone due to limited resolution of seismic tomography and sparse distribution of geodetic stations. We have rewritten this part in the revised manuscript.

1.17: Line 56: Whether it is easy or not is subjective. What are the authors really trying to say here? Should be rephrased. Is there an simple way to explain what “downward continued to the seafloor” means for readers who are not seismic experts?

Response:[p. 3] As noted by Arnulf et al. (2018), the sources and receivers for multichannel seismic data are located on the sea surface, far from the seafloor. The refractions, which carry critical velocity information, are typically masked behind strong scatterings from the seafloor, preventing a tomographic approach. Downward continuing multichannel data simulates a seismic acquisition right above the seafloor and produces a new dataset with virtual sources and receivers near the seafloor. In the new data, crustal refracted arrivals become first arrivals that can be easily used for tomographic imaging. We have added more description about downward continued data to make it easily to follow for readers who are not familiar with seismic acquisition.

1.18: Line 58: You should note that it was called the “SMR” in Arnulf et al. (2018), ref #9 here, and that you are calling it something different here (“EMR”).

Response:[p. 2] We have rephrased the abbreviation of the secondary magma reservoir as “SMR” here and in Fig. 1.

1.19: Line 64: In the caption for Figure S1, explain what Q_p is and how it relates to the figure. Also can you add a plain language explanation of what “viscoacoustic” means in this context?

Response:[Fig. S1, p. 4] Q_p is the P-wave quality factor to characterize seismic wave attenuation, which is defined as $Q_p = 2\pi \frac{E}{\Delta E}$, where E is the elastic energy in a wave cycle and ΔE is the dissipated energy in a cycle. Small Q_p values indicate strong attenuation, while large Q_p values reflect weak attenuation. We have added more explanation in the caption of Fig. S1. In perfectly elastic solids, the stress is proportional to the strain but independent of the rate of strain, and the mechanical energy is stored without dissipation. On the other hand, for perfectly viscous liquids, the stress is directly proportional to the rate of strain and independent of the strain itself, but in this case the energy is completely dissipated (Toksöz et al., 1988; Carcione, 1993). A realistic representation of the Earth can combine the mechanical properties of elastic solids and viscous liquids. In the resulting material, the stress depends upon both the strain and the rate of strain. Such a medium that has the characteristics of solids and liquids is named viscoelastic. A similar anelastic medium but only allowing P-wave propagation is called viscoacoustic, and the corresponding wave equation is viscoacoustic wave equation. In the main text, we have added a plain language explanation for attenuation and viscoacoustic.

1.20: Line 65: Cross-sections of “attenuation” are important in the rest of the paper. I think it would help to take a sentence or two here to explain what “attenuation” means in this context and what these cross-sections show.

Response:[p. 4] Seismic attenuation is a metric of elastic wave energy dissipation (Toksöz et al., 1988; Carcione, 1993). It strongly depends on temperature, partial melting and water content, and can provide valuable information on Earth’s internal structure and dynamics, in complement to that learned from seismic velocity (Karato and Jung, 1998; Takei, 2017; Debayle et al. 2020). We have added more explanation in the updated manuscript.

1.21: Line 72: What data are fit better? Need to better explain this.

Response:[p. 4] Comparisons of synthetic and observed data show that the estimated velocity and attenuation models from the FWI can produce much more accurate traveltimes and waveforms than those calculated using the initial tomographic models. In addition, the incorporation of attenuation in the FWI enables us to match the weak large-offset amplitudes.

1.22: Line 75: How much of a velocity anomaly or how low a low-Q anomaly? And at what depths? Is this claim substantiated in the figures in the Supplementary materials? If so, which figures?

Response:[p. 4] The descriptions in the original manuscript are not clear. We have revised this sentence to make it easier to understand in the updated manuscript.

1.23: Lines 82-84: To me, this sentence is poorly constrained speculation. “Hydrothermal cooling above the MMR” is a pretty generic explanation and is unconvincing. I can think of other interpretations (the existence of thick, dense, ponded lava flows in the subsiding caldera floor block, for example), but I don’t think there is compelling evidence for any one idea. Likewise, the low-velocity boundaries could be just due to the caldera faults. There is nothing that convincingly identifies them as “magma pathways”. In fact, I might expect a zone of diking to have higher velocities. Then there is the problem that zones of diking are likely vertical and narrow, so would be hard to image seismically. Also, the references cited in this sentence seem generic and do not particularly substantiate the interpretations. For all these reasons, I would omit this sentence.

Response:[p. 4] We agree with the reviewer’s interpretation. The high-velocity core could be formed by thick, dense, ponded lava flows on the subsiding caldera floor, and its low-velocity rims are more like to be the inward-dipping fault zones formed by the subsidence of the caldera floor. If the diking zones are vertical stripes, it is very difficult to image using reflection data because we cannot record relative reflection signals at the surface (see the figure in the response to question 1.4). We have rewritten these two sentences in the updated manuscript.

1.24: Line 85: Does attenuation (Q_p) have units?

Response: The quality factor (Q_p) is defined as $Q_p = 2\pi \frac{E}{\Delta E}$, where E denotes the total energy in one wave cycle and ΔE denotes the dissipated energy in one cycle. Therefore, Q_p is dimensionless, i.e., having no physical units.

1.25: Lines 86-88: How does hydrothermal alteration “induce” high-porosity? Not obvious to me. Cracking itself and deformation make high porosity. Or the extrusive layer itself starts with high porosity.

Response:[p. 4] We agree with the reviewer on this point. After reviewing additional references, we have identified that off-axis, cool, downflowing fluids can induce cracking in the host rock, thereby increasing porosity. However, near the axis, ascending hot fluids are likely to heat the surrounding rocks and potentially reduce preexisting porosity through thermal expansion or rapid deposition of hydrothermal minerals. Therefore, the observed high porosity may be associated with the extrusive basaltic layer, as well as cracking and deformation caused by Axial volcano inflation and deflation, and velocity heterogeneities. This sentence has been revised in the updated manuscript to reflect these considerations.

1.26: Line 95 (Figure 2): In Fig. 2, I would change the y-axis labels to “depth below sealevel” to clarify that it is not depth below the seafloor. What is the “perturbed V_p model” in Figure 2c? And what is the difference between Fig. 2a and c? This needs some explanation if this is what you are referring to as what is showing “more detail”. Is it just imaging the MMR in more detail? Or is it imaging multiple magma reservoirs in the shallow crust in more detail? Also, I think the blue contours in Figure 2c deserve some explanation in the text. What is interpreted that this contour signifies and what is it based on? I would also add a plain language explanation of what Fig. 2b shows and why it makes intuitive sense (if it does). What is “seismic attenuation” exactly, and why isn’t there major (and more uniform) attenuation inside the MMR?

Response:[Fig. 2, p. 5] We have changed the y-axis labels to “depth below sealevel” in Fig. 2. The perturbed V_p ($\delta \ln V_p$) model is the relative P-wave velocity anomaly, and it is calculated as $\delta \ln V_p = (V_p - V_{p0})/V_{p0}$ (detailed calculation is shown in the Methods), where V_p is the estimated velocity model (Fig. 2b) from FWI and V_{p0} is the background velocity model (Supplementary Fig. S1c). The relative velocity anomaly model ($\delta \ln V_p$) shows more details for multiple magma reservoirs than the absolute velocity model (V_p). The blue contours are plotted according to the different melt fractions estimated in Fig. 4b. It shows that the melt fractions are zoned in the MMR and underlying conduit. We have added more explanations in the figure caption and main text. Seismic attenuation reflects the energy dissipation during seismic wave propagations, which can be caused by viscous friction across grains and dislocations, fluid motions in pores and fractures, as well as wave scattering due to velocity heterogeneities. From our FWI model, we see that the melt fraction in the upper MMR is greater than 32%, which is a framework of fluid-suspended crystals and has significantly reduced viscosity. However, in the lower MMR, the melt fraction decreases, leading to increased viscosity and low Q values. To enhance the understanding of Fig. 2b, we have added more plain-language explanations in the updated manuscript.

1.27: Line 96: You need to indicate which panel in Figure 2 the reader should refer to to see this. Is this the red/white “smile” at 2km depth in Fig. 2c? Also is the “P-wave speed anomaly” fast or slow? And is it really “at the top” of the MMR, or above it?

Response:[Fig 2, p. 5] The thermal boundary layer is the “smile” at ~ 1.5 km below the seafloor as shown in Fig 2c. We have added a marker in Fig 2c to show its location. The P-wave speed anomaly is slow. We have revise this sentence in the update manuscript. In Arnulf et al. (2018)’s tomography study, the MMR is identified at the location shown by the red polygons in Figs. 2a and 2b. Our FWI produces a higher resolution result for the MMR (Fig. 2c), with its roof is slightly deeper than Arnulf et al. (2018)’s result. In our FWI models, the slowest P-wave anomaly is observed in the upper MMR, as depicted in the newly plotted Fig. 2c.

1.28: Line 98: Wait a minute ... there is “high-melt fraction” in the pink “smile” at 500 m depth below the caldera? I thought the magma was in the MMR at ~ 1 km depth below the seafloor. This needs clarification.

Response:[p. 5] We agree with the reviewer on that the magma is in the MMR at depth ~ 1.6 km below the seafloor. The pink “smile” at 500 m depth below the caldera in the original manuscript is not the magma reservoir. We have added more labels on Fig. 2c to show the MMR and clarified in the text that the partial melt is in the upper MMR.

1.29: Line 103: What does this mean or imply and how does it make sense?

Response:[p. 5] The description in the original manuscript may not be clear. As in the response to question 1.27, we observed that the lower MMR has a smaller Q value than the upper MMR. This difference may be attributed to variations in melt fractions. In the upper MMR, the melt fraction is near the eruption threshold ($\sim 30\text{-}50\%$), resulting in a framework of fluid-suspended crystals. This leads to decreased viscosity and relatively higher Q values. In contrast, the lower MMR has a smaller melt fraction and is in a crystal-supported framework, resulting in increased viscosity and smaller Q values. We have revised this sentence in the updated manuscript to provide a clearer explanation.

1.30: Line 108: I would say ~ 7 km east of the MMR, if you measure from the center of the MMR to the center of the EMR.

Response:[p. 5] We have revised this sentence according to the reviewer's suggestion using the distance from the center of the MMR to the center of the EMR.

1.31: Lines 111-112: Depth below sealevel? Or depth below the seafloor?

Response:[p. 6] In the original manuscript, the measure of the depth is below the sea level. In the updated manuscript, we use a depth measure below the seafloor (bsf) according to the reviewer's suggestion.

1.32: Line 117: Can this method really identify something of this scale at this depth? How do you know this is actually a fluid conduit?

Response:[p. 6] In seismology, the Rayleigh criterion is commonly used to measure the resolution of seismic imaging, which states the resolution is approximate one fourth wavelength. Inversion-based methods, such as least-squares migration and full-waveform inversion, offer higher resolution than one fourth wavelength because they reduce the Hessian blurring effect. The P-wave velocity in the magma reservoir is around 4.6 km/s, and the highest frequency used in the full-waveform inversion is 9Hz. Therefore, the resolution is 127 m or less. From our FWI model, we observed the channel between MMR and WMR is greater than 200 m. Theoretically, it can be resolved using the FWI method. Based on the velocity, temperature, and partial melt models, we observed that this channel exhibits low velocity, high temperature and $\sim 20\%$ melt fraction. Therefore, it may be a magma conduit connecting the MMR and WMR.

1.33: Line 118: I can't help but wonder how confident can we be of this interpretation... This is just one 2D seismic line. Would these "extra" low-V zones interpreted as magma reservoirs be imaged on other lines from the 2002 MCS survey? The evidence would be stronger if you saw the same things on other 2D lines and that made sense in three dimensions.

Response:[p. 6] At the current stage, we do not have access to more MCS data to constrain the detailed 3D structure of the MMR and WMR. For the data of the J48 survey line, we have done many tests of FWI by adjusting different parameters, including frequency bands, gradient smoothing size and optimization methods. All tests showed that the low-velocity anomaly of WMR is a stable feature. We have also carefully checked the 3D tomography model from Arnulf et al. (2018). The velocity profiles of the J54 and J63 survey lines, which run parallel to the J48 line, show the continuity of the low-velocity anomaly in the MMR towards the west-southwest direction (see the white arrows on the following J54 and J63 velocity profiles). In addition, the velocity profile of the J45 line, which is perpendicular to the J48 line, reveals a low-velocity anomaly at a depth of 2.5 km below the sea level (see the dashed circle on the J45 velocity profiles). Although the tomography resolution is limited, these low-velocity features are co-located with the WMR imaged in our FWI model, providing evidence supporting our identification of the WMR. In addition, we also examined the reflection images from Carbotte et al. (2020). We observed strong reflection events at the top of the WMR, MMR, SSMR, and SMR as shown in the following figure. Previous studies did not interpret these reflections in detail, but they are co-located with our newly identified multiple magma reservoirs in the shallow crust (see the red arrows and labels in the following reflection image).

[Figure redacted]

1.34: Lines 118-121: I would say that the distribution of seismicity on the caldera ring fault doesn't necessarily say anything about whether the WMR could feed a dike to the surface in the western part of the caldera. That said, there aren't many eruptive fissures there from the last several hundred years from the mapping of Clague et al. (2013). In general I think it's dubious that MCS methods can image narrow vertical structures like dikes in the shallow crust. I would not expect to see them, even if they do exist.

Response:[p. 6] We agree with the reviewer that there aren't many eruptive fissures on the seafloor in the western part of the caldera. If the stresses on the western and eastern caldera shoulders are similar during volcano inflations and deflations, and if there are magma pathways from deep high-melt reservoirs to their seafloor, eruptions can occur on both sides of the caldera. However, actual eruptions only occur from the eastern side. In this study, we aim to reveal the subsurface volcano structure and try to explain why the magma does not transport from subsurface magma reservoir to western caldera. From our FWI models, we observed that there is no low-velocity zone beneath the western caldera, but there is a small WMR, which helps in understanding volcano structures beneath Axial Seamount. Regarding seismic imaging for vertical dikes, the reflection method may not work because effective signals reflected from the vertical dikes cannot be received on the surface (see the response to question 1.4). However, the FWI method uses not only the reflections but also refractions, diving waves and scatterings, thus having the potential to image the vertical dikes. Of course, the resolution depends on the frequency and optimized strategies used in the FWI, and the imaging accuracy for thin vertical dikes using FWI needs further systematic investigation.

1.35: Lines 121-124: I'm equally dubious of this interpretation. I'm not convinced this "LV throat" is real or significant. The seismicity shows slip along the caldera faults during inflationary or deflationary deformation. It doesn't necessarily say anything about diking or the existence of a magma pathway there. In fact, one might expect a "diking pathway" to have a high velocity. I'm not convinced that MCS can image narrow vertical structures like "dike pathways".

Response:[p. 6] Seismic velocity is determined by bulk modulus (K), shear modulus (G) and density (ρ). If magma intrudes into sedimentary rock, the dike should exhibit a higher P-wave velocity than the host rock because the igneous rock commonly has greater bulk modulus and density. But beneath Axial Seamount, the caldera was formed by historical intrusion and eruption events. The host rock is old and denser, whereas the dike is new and lighter. The diking pathway may have a slower velocity than surrounding rock. As for the accuracy of imaging the LV-throat, as discussed in response to question 1.34, it is difficult for reflection imaging methods to image narrow vertical dike pathway. However, FWI uses all kinds of wave arrivals, including reflections, refractions, diving waves and scatterings, and has potential to image fine-scale features. The feasibility of FWI has been demonstrated, such as imaging detailed gas reservoir structures in the North Sea (Operto et al., 2015, Geophysical Journal International) and complex salt structures in the Gulf of Mexico (Wang et al. 2019, The Leading Edge).

1.36: Line 141: I wonder why there isn't a high melt fraction at the very top (roof) of the MMR in Figure 4b. Isn't that where the MCS data show the highest seismic reflections indicative of a change from solid to melt?

Response: The blue polygon in original Fig. 4b outlines the location of the MMR identified by Arnulf et al. (2018). They defined the location according to reflection images. Because the migration velocity is built using tomography, the image uncertainty is relatively large. Based on our FWI models, the roof of the MMR is slightly deeper than Arnulf et al. (2018)'s model. Furthermore, our model shows that the top of the MMR (named the upper MMR in the text) has the largest partial melt, approximately reaching the eruption threshold.

1.37: Line 143: What you are calling the “upper MMR” in Figure 4, looks like the middle of the MMR to me, based on the pink and blue outlines.

Response:[Figs. 2 and 4] In original manuscript, the pink and blue polygons in Fig. 4 are the MMR location identified by Arnulf et al. (2018). In our new FWI model, the roof of the MMR is slight deeper. We have replotted Figures 2 and 4 to show the locations of upper and lower MMR based on our FWI models.

1.38: Line 145: Is the volume of melt in the “upper MMR” a subset? If so, what percent of the MMR is the “upper MMR”? How can you constrain the volume of the “upper MMR”, based on only one 2D MCS line? How do you know the 3rd dimension for calculating a volume? Needs clarification.

Response:[p. 7, Fig. S21] We estimate the melt volume by combining our partial melt model and the 3D MMR geometry extracted from the tomography model of Arnulf et al. (2018). The 3D geometries of the whole MMR and high-melt upper MMR are shown in Fig. S21. The melt volume in the high-melt upper MMR is a subset of the whole MMR, and it is about 4-6% of the whole MMR melt according to our estimation.

1.39: Line 147: Note reference #11 is duplicated as reference #43.

Response: We have removed the original reference of #43.

1.40: Line 150: It's still unclear what constitutes the “upper” vs. “lower” MMR.

Response:[Figs 2c, 4b, p. 7] We have added labels on Figs. 2c and 4b to show the locations of the upper and lower MMR. They are identified according to the estimated partial melt fractions.

1.41: Line 162: What is the “boundary layer” in Figure 5?

Response:[Fig. 7, 8] Here the “boundary layer” means a thermally controlled permeability boundary located at the base of the upper crustal dike, below which rising melt pond and accumulate, forming the axial magma reservoir (Suzanne et al., 2021). We have modified the label of original Fig. 5 and added more explanation in the caption.

1.42: Line 168-170: I’m not convinced the data show this, especially the “LV throat”. You could have drawn these features on Figure 5 before this study was done, because it was already obvious there is some sort of focusing of magma from the top of the MMR into dikes that intrude upward near the eastern caldera fault, but I’d say the reasons for that are still unclear. To me, the results in this paper do not illuminate why that is the case.

Response:[p. 8] Previous studies, including seismicity, active- and passive-source tomography, seismic reflection imaging, and multibeam mapping, have revealed the geometry of the MMR and seafloor fissures, but they do not provide high-resolution subsurface structure beneath Axial Seamount. In this study, we utilized the advanced full-waveform inversion method to constrain P-wave velocity and attenuation models of the J48 survey line, followed by calculating temperatures and partial melt fractions. These models revealed multiple magma reservoirs in the shallow crust and a magma pathway beneath the eastern caldera, which improves our understanding of the subsurface structure of this marine volcano and provides geophysical evidence for previous geological hypotheses.

1.43: Lines 173-178: This explanation for the difference between the eastern and western sides of the caldera seems very speculative.

Response:[p. 8] Previous seismicity studies show that more earthquakes occurred on the eastern outward-dipping fault than on the western fault, and the earthquake sources terminate deeper on the eastern wall than on the western wall. In addition, using vertical deformation data, Hefner et al. (2020) demonstrated that the 2015 eruption can be approximated by a nearly vertical prolate-spheroid pressure source, but the eastern caldera fault has the most slip during the eruption. This suggests that the pressure in the magma conduit of the middle-to-low crust is relatively equal for both sides, but the stresses exerted on the caldera walls in the shallow crust are different. Here, combined with our FWI models, we attempt to provide an explanation for the differential stresses in the shallow structure of Axial volcano.

1.44: Line 216: Actually, Clague et al. now think the present caldera only formed about 1100 years ago (the age reported in Clague et al. (2013) is incorrect). This was presented in an AGU abstract: Clague, D. A., R. A. Portner, J. B. Paduan, M. Le Saout, and B. M. Dreyer (2019), Formation of the summit caldera at Axial Seamount. Abstract presented at 2019 Fall Meeting, AGU, San Francisco, CA, 9-13 Dec.

Response:[p. 10] We have noted this AGU report and updated the formation time of the caldera in the manuscript according to Clague et al. (2019).

1.45: Lines 223-225: This is an interesting idea, but what would explain the SMR?

Response: The secondary magma reservoir (SMR) is located to the east-northeast of the MMR. The J48 survey line does not cross the center of the SMR. Therefore, based on this study alone, we cannot accurately infer the formation of SMR. Arnulf et al. (2014, 2018) have suggested possible magma conduits linking the MMR and SMR based on seismic migration sections. To gain a better understanding of magma accumulation processes within the SMR and magma movement between these two reservoirs, more geophysical data are needed to be acquired over the SMR.

1.46: Lines 229-232: This is the first acknowledgement that all the results in this paper are based on only one 2D seismic line and that this could limit what you can conclude! In my opinion, there should be much more of this type of discussion of the limitation of this study, and what would be needed to better substantiate the tentative results presented.

Response:[p. 6, 10] We have added more descriptions in the revised manuscript about the limitations of the 2D seismic investigation.

1.47: Another important question that should be addressed in the paper is: if they are real, why were the WMR and ESMR (the 2 “extra” magma reservoirs) not previously identified by seismic reflection methods in the 2002 (and 2019) MCS surveys? To me, seismic reflection is a more compelling way to identify where magma bodies exist in the crust, because of the strong reflections produced by the change from solid rock to liquid, whereas mapping seismic velocities seems like a less definitive and more indirect way of doing the same thing. Why should readers believe these new results based on only seismic velocities if they are not confirmed by reflection seismology?

Response: We have carefully checked the reflection images calculated using reverse-time migration from Carbotte et al. (2020). As shown in the following figure, there are actually some coherent reflection events at the top of WMR, SSMR and SMR identified in this study. But previously studies did not interpret these reflection energies in detail. The co-locations of these coherent reflections provide evidence for our newly identified magma reservoirs.

[Figure redacted]

II. Responses for Reviewer 2

2.1: My suggestion for a minor revision concerns the discussion of melt fraction. I confess that I have a hard time wrapping my mind about quantifications like “ranging from 39% to 65%” (line 144). First, melt fraction is likely a matter of spatial scale. I presume that at the finest scale, the maximum is 100%, meaning that somewhere in the magma system is a little pocket of pure melt. So, a number like 39% is only meaningful in the sense of representing an average with respect to some volume of material (rock+melt). As the imaging is tomographic, I suppose that the relevant averaging volume ought to be based on the volume of the resolving kernel. The Supplementary Material mentions resolution in connection with Figs 10-12, but I was unable to discern how to use them to figure out this volume. After pondering the issue for a while, I would like to make the suggestion that the authors add an “exceedance plot” like this:

[Figure redacted]

Here, X is informed by some estimate of the resolving power of the tomography. Such a cumulative histogram would provide more information than what's in line 144. It could be generated very easily from the model underpinning the authors Fig 4b.

Response:[Fig. S20] We agree with the reviewer that the estimated melt fraction represents an average with respect to the inclusions of solid basalt and basaltic melt. According to the Rayleigh criterion, the resolution of seismic imaging is approximately one fourth of the seismic wavelength. Therefore, the resolving kernel is about 83 m at a shallow layer with $V_p \sim 3$ km/s and is around 167 m at a deep layer with $V_p \sim 6$ km/s. Following the reviewer’s suggestion, we have added an exceedance figure in the support information (Fig. S20) to show the relation between the melt volume and partial melt fractions (refer to the following figure).

The error bars are based on two end-member partial models: one with spherical inclusion and the other with a vertically aligned elliptical body with an aspect ratio of 0.1. The fractions falling between the two dashed red lines represent the potential maximum partial melt range.

2.2: I also confess that ranges like “39% to 65” (line 144) raise more questions with me than they answer. Is the range estimate a confidence interval, e.g., 52% +/- 13? If so, what is the formal confidence? Is it 95%, or something else? What sources of variation contribute to this confidence, and has “model error been included. I imagine that model error (including the ability to map V_p and Q_p into melt fraction) is the main source of error. But speaking only for myself, and not for the authors, I have no confidence in my ability to quantify such an error. Anyway, I think that the authors ought to explain what they intend a range to mean. Such an explanation could be based on the exceedance plot (as in my figure). I also wonder whether an intercomparison of a range with that stated in another paper (as the authors provide, e.g., in line 40) is really as straightforward as it seems. This point could be addressed with a few additional lines of text.

Response:[p. 21] We agree with the reviewer that there is a range of confidence interval and the confidence depends on many factors. In this study, we first explained the low-velocity anomaly as much as possible using temperature variations, and then estimated the partial melt fractions with temperature above 1,150 °C. Different inclusions have been tested to calculate melt models, and the models with spherical inclusion and vertically aligned elliptical body with an aspect ratio of 0.1 are used as the two end members. The above exceedance plot has shown the error bars constrained from the two end-member models. We have added more explanation about the influence factors on the accuracy of estimated partial melts in the updated manuscript.

III. Responses for Reviewer 3

3.1: In discussing the connection between magma bodies, the authors seem to be inferring 3-D geometry from a 2-D inversion. A 2-D inversion can show bodies are connected but not demonstrate that they are not. It is impressive that the inversions can image the magma pathway on the eastern wall, but we know from previous studies that is where eruptions occur because that is where the plate boundary is, so what is new in the interpretation?

Response: In this study, we utilized the advanced full-waveform inversion method to constrain P-wave velocity and attenuation models along the J48 survey line, followed by detailed analysis of temperature and partial melt structures. Because full-waveform inversion can utilize not only reflections but also refractions, diving waves, and scattering, it can resolve very fine-scale features. Its feasibility has been verified by many successful applications in seismology, such as providing high-resolution images for gas reservoirs in the North Sea and revealing complex salt structures in the Gulf of Mexico. Some descriptions about the connection between magma bodies in the original manuscript may be inaccurate. We have rewritten and revised them in the updated manuscript. Previous studies, including multi-beam mapping, seismic tomography, and earthquake location, have shown the geometry of the main magma reservoir (MMR) beneath Axial Seamount and the location of plate boundary along the eastern wall. However, they do not reveal the detailed structure below the surface fissures or how these fissures connect with the high-melt magma reservoirs at depth. This study presents high-resolution P-wave velocity models, attenuation models, temperature profiles, and partial melt distributions, which clearly map the connection between the MMR and surface eruptive fissures (plate boundary). Therefore, the innovation of this paper includes: (1) high-resolution P-wave velocity and attenuation models along the J48 survey line, (2) temperature and partial melt profiles, (3) multiple magma reservoirs in the shallow crust beneath Axial Seamount, and (4) connections between subsurface magma reservoirs and between these magma reservoirs and surface fissures. Some additional details can also be found in the response to question 1.1.

3.2: The paper does develop some ideas about the asymmetry of the magma chamber relative to the caldera and its influence on the location of eruptions. There is also speculation that the connectivity to the western magma body may play a role in supporting asymmetric on the caldera fault and several short deflation events observed from 2016 to 2019, but the manuscript does not make any attempt to test this idea with the geodetic data. With some thought and more reference to the prior literature, I think this manuscript could be revised into an inciteful publication.

Response:[p. 9] We agree with the reviewer that geodetic analysis will help to verify the model proposed in this study. We carefully reviewed the work of Chadwick et al. (2022) and found that during the June 2018 short-term deflation event, the bottom pressure recorder on the western caldera wall experienced greater vertical displacement than those on the eastern wall (see the following figure). This may indicate that it is more possible that magma movement beneath the western wall, such as from the MMR to WMR, led to this deflation event. We also provide a detailed explanation in response to question 3.19.

[Figure redacted]

3.3: 14-51. It is not clear what questions are motivating this study?

Response: According to three reviewers' comments and suggestion, we have carefully revised our manuscript, rewriting many places. Previous studies have revealed the location and geometries of the main magma reservoir, as well as the surface eruptive fissures. However, few studies have explored whether additional fine-scale magma reservoirs exist beneath the caldera, and there are no high-resolution velocity and attenuation images of Axial Seamount directly showing the conduit from the MMR to surface fissures. Our study, based on advanced full waveform inversion, identified multiple magma reservoirs in the shallow crust, in addition to the MMR, and revealed a possible magma pathway beneath the eastern wall. This provides important geophysical evidence to reveal the magma plumbing system of Axial Seamount. In addition, we constrained the temperature and partial melt profiles based on the velocity and attenuation models, which are more accurate than those using only velocity models. A high-melt zone in the top layer of the MMR is observed, with a melt fraction near the eruption threshold and a comparable volume to recent magmatic events. This indicates it may be the magma source of recent eruptions.

3.4: 34-36 "Although previous studies revealed the location of the subcaldera magma chamber, it remains unclear why all recent eruptions were only from the eastern wall rather than the two seismically active walls." I disagree, it is very clear why recent eruptions are on the east wall. This is where the plate boundary is and where the summit connects to the north and south rifts, one of which also spreads in each eruption and takes most of the melt volume. Given the location of the plate boundary (e.g., Chadwick et al., 2013), how can a spreading event be supported by an eruption on the west wall? It is interesting that the asymmetric offset of the main magma chamber relative to the caldera which was first reported by Arnulf et al. (2014) centers it beneath the plate boundary.

Response:[p. 2] We agree with the reviewer that the plate boundary is along the eastern caldera wall. Chadwick et al. (2013, 2017) have reported the eruptive fissures associated with recent eruptions. According to the first reviewer’s suggestion, we have rewritten this sentence in the updated manuscript. This work identifies multiple magma reservoirs in the shallow crust, in addition to the main magma reservoir documented in previous research. Furthermore, the high-resolution images generated through full waveform inversion reveal a low-velocity throat beneath the eastern caldera wall, potentially serving as a magma pathway from the high-melt reservoir to surface fissures. These findings augment our understanding of the subsurface magma plumbing structure beneath Axial Seamount and provide geophysical evidence supporting the eastern offset of recent eruptions.

3.5: 48-50. “It is unclear whether there is a high-melt zone beneath the caldera with a melt fraction close to the eruption threshold and a volume comparable to the size of recent magmatic events.” If it is unclear what is the alternative? This is the only mechanism that I am aware of to explain eruptions on mid-ocean ridges and the presence of a shallow high-melt zone supporting eruptions has been demonstrated along the East Pacific Rise (e.g., Singh et al., 1998; Xu et al., 2022) and the Juan de Fuca Ridge (Canales et al., 2006).

Response:[p. 3] According to the first reviewer’s suggestion, we have rewritten this sentence in the updated manuscript. In the revision, we emphasize the partial melt distribution in magma reservoirs and the connection between the high-melt zone in the MMR to the focusing of eruptive fissures near the eastern caldera wall in 1998, 2011 and 2015.

3.6: 79-80. What is the significance of the shallow high velocity body in the caldera? At least in some other calderas, the caldera infill has a very low velocity (e.g., Hooft et al., 2019). I think a review of prior observations in calderas would be useful to interpret this result which is presently not interpreted.

Response:[p. 4] We have carefully reviewed the reference of Hooft et al. (2019). A 3-km-wide, cylindrical low-velocity anomaly in the upper 3 km was revealed beneath the north-central portion of the Santorini caldera. Hooft et al. (2019) interpret this anomaly as a low-density volume caused by excess porosities up to 28%, with pore spaces filled with hot seawater. As suggested by the first reviewer, the high-velocity core in the caldera may be formed by thick, dense, ponded lava flows on the subsiding caldera floor. We have rewritten this sentence in the updated manuscript.

3.7: 80. “246 m” - To this level of precision?

Response: In seismology, the Rayleigh criterion (one fourth wavelength of the seismic wave) is commonly used to measure the resolution of seismic imaging. FWI can produce a better resolution because it iteratively reduces the blurring effect of the Hessian matrix. Assuming a P-wave velocity of 3 km/s at the caldera and the maximum frequency of 9 Hz used in FWI, the wavelength is ~ 333 m and the scale that FWI can resolve is at least 83 m. Therefore, for a 200-m thick anomaly, FWI can provide a good constraint.

3.8: 81-82. The inward dipping normal faults are mainly identified by the caldera walls. Most of the seismicity is on the buried outward dipping ring fault. While both Arnulf et al. (2018) and Waldhauser et al. (2020) associate some earthquakes with the inward dipping fault on the eastern side of the caldera, the catalogs are inconsistent.

Response:[p. 4] We agree with the reviewer that the inward-dipping faults are mainly constrained by caldera walls. While some seismicity studies can detect these faults, there may be variations in their catalogs. We have revised this sentence in the updated manuscript.

3.9: 82-84. The first half of this sentence seems to be referring to structure right above the magma body which doesn't seem to be mentioned anywhere else. The second half seem to be referring to the feature that is presented in more detail in line 121-124. The whole sentence is orphaned in a section on shallow structure and thus quite misplaced.

Response:[p. 4] According to the suggestion of the first reviewer, we have rewritten these two sentences to explain the formation of the high-velocity caldera floor.

3.10: 110-114. Since the center of the SMR is well to the south of the MMR (Arnulf et al., 2014) and this profile, there is no justification for concluding it lacks connectivity to the MMR from a 2-D model. I think this is a major flaw in the interpretation.

Response:[p. 6, 10] We agree with the review about this point. The SMR is to the south of the MMR and the J48 survey line does not cross the center of the SMR. It is difficult to infer the connection of the MMR and SMR in real 3D structure. We have revised this sentence to clarify that the SMR may not connect to the MMR along the image of the J48 line. In addition, we cannot rule out intermittent connection zones between the MMR and SMR that locate out of the image plane of the J48 survey line.

3.11: 117. Is the WMR seen in the reflection data? If not, why not given the sharp velocity gradients near its top?

Response: The following figure is the reflection image calculated by Carbotte et al. (2020). As in the response to question 1.47, there are actually some coherent reflection events around the newly identified WMR and SSMR. But previous studies did not interpret these reflection energy in detail.

[Figure redacted]

3.12: 119-121. I do not follow. Geometrically, the fault must terminate in a mushy zone that can deform viscously, so this could be a magma conduit. From the plots in Figure 4b and S15-18, it looks suspiciously like the earthquakes do root in the combined melt body, producing a notch in it, potentially because of enhanced cooling.

Response:[p. 6] We agree with the reviewer that the fault should terminate in a ductile/viscous zone. Based on seismicity distribution, there are some earthquakes in the magma conduit between the MMR and WMR, but their number is not as large as that beneath the eastern caldera wall. As suggested by the reviewer, we have revised this sentence in the update manuscript to ensure accuracy.

3.13: 121. "In addition". Should this be "Second" based on lines 114-115.

Response:[p. 6] We have changed "In addition" to "Second" in the updated manuscript.

3.14: 122-124. How does this pathway compare with those inferred by Arnulf et al. (2014) from a 3-D inversion?

Response: In Arnulf et al. (2014), multiple 2D lines from the 2002 survey, instead of real 3D inversion, are used to image the structures of MMR. We have carefully analyzed the pathways proposed by Arnulf et al. (2014). As shown by the red arrows in the following figure, there are a couple of dipping pathways in the shallow crust that link the MMR and seafloor fissures along the reflection profile of the J48 line. The low-velocity throat identified in this study may be the projection of these pathways on the velocity image of the J48 line.

[Figure redacted]

3.15: 145. How do the authors get this estimate? In Figures 4b/S15 the 32% region looks to be 0.2 km thick and 3 km wide so for the upper limit of 0.57 km^3 , this requires a 3rd dimension of $0.57 / (3 \times 0.2 \times 0.32) = 3 \text{ km}$ but the magma chamber extends 15 km along axis. If the 26% contour was used, there would be much more melt, so why select the 32% contour other than to get the answer that is desired? Some clarification in supplementary docs would be helpful.

Response:[p. 7] We estimate the melt volume by combining our partial melt fraction model with the 3D geometry of the MMR described in Arnulf et al. (2018). As shown by Arnulf et al. (2014) and Chadwick et al. (2016) (see the following 2D zoned MMR figure), the MMR is zoned with different melt properties. The region just below the caldera exhibits strong reflection amplitudes, indicating a high melt fraction, while the northern region shows weak reflection energy, suggesting a smaller melt fraction. Therefore, the high-melt upper MMR does not extend to the entire MMR; it is only located just below the caldera (see the following 3D MMR figure). We calculate the volume by first applying an area integral over the upper MMR based on our melt model and then using the 3D MMR model to constrain the length. The total melt volume is calculated similarly. We have added more description about the volume calculation in the main text.

[Figure redacted]

3.16: 156-159. How does this interpretation relate to previous interpretations of magma chambers on mid-ocean ridges? Isn't this the long-standing interpretation of crustal magma chambers at mid ocean ridges (Sinton and Detrick, 1992; Singh et al., 1998)?

Response:[p. 8] We have carefully reviewed the papers by Sinton and Detrick (1992) and Singh et al. (1998). The crustal magma chambers at the mid-ocean ridge consist of melt and mush with varying partial melt fractions. Here, our aim is to reveal the detailed melt structure of shallow magma reservoirs below Axial Seamount. We have revised this sentence and cited these two references in the updated manuscript.

3.17: 173-176. If magma transport in and out of the WMR relaxes stresses why are there remaining stresses? Hefner et al. (2020) have modeled caldera inflation with asymmetric fault motions and show the remaining caldera inflation is quite symmetric, so I am not sure what need to be explained by transport of magma in and out of the WMR? The fault motions are asymmetric because the center of inflating magma body is offset to the east of the caldera.

Response:[p. 8] Hefner et al. (2020) inverted a nearly vertical spheroid source to model caldera inflation during the 2015 eruption, with a depth of 3.01 km, by removing fault-induced deformation. As analyzed by Hefner et al. (2020), the source pattern indicates that the pressure in the magma conduit below the MMR is symmetric. However, the different slips on the western and eastern faults indicate that the stress in the shallow crust is not symmetric. Our FWI models can provide some insight to explain such asymmetric stress results in the shallow crust. The WMR may accommodate a part of the pressure beneath the western wall, and the viscous magma infilling and movement in the MMR and WMR may still produce some residual pressure. This part of the pressure can be accommodated by the slip on the western outward-dipping fault plane.”

3.18: 178-180. This is one reasonable explanation and that adopted by Hefner et al. (2020), but an alternative explanation is that the west wall is less coupled.

Response: Factors controlling fault coupling include a combination of effective stress and frictional properties related to rheological and geometrical changes on the fault interface. Faults with low coupling accommodate most of their slip as stable sliding, while the slip on faults with strong coupling is commonly accommodated by large earthquakes. If the western wall is less coupled than the eastern wall, the western outward-dipping fault can be stably sliding and generate fewer earthquakes. Despite less seismicity recorded at the surface, its slip should be the same as the eastern wall if the pressure is the same on both sides. However, Hefner et al. (2020) have shown that during the 2015 eruption, the eastern fault had the most slip (with a maximum of 3.6 m), while the western fault had a maximum slip of 0.8 m. Therefore, it is challenging to use low coupling of the western fault to explain its different seismic activity compared to the eastern wall. Of course, accurate fault coupling analysis of the two faults and related factors requires future detailed investigation.

[Figure redacted]

3.19: 180.-182. The short-term deflation events from 2016-2019 are intriguing and loss of magma to the WMR is a feasible explanation, but such short-term deflation events were not seen between the 1998 and 2011 eruptions and the 2011 and 2015 eruptions so they are not related to a process that is inherent to the eruptive cycle as the model presented here seems to suggest. If these events are related to loss of magma into the WMR, then they should be geodetically observable by comparing the pressure gauge and tilt meter from the cabled instruments at the ASHES vent field on the western side of the caldera with those elsewhere. Have the authors looked?

Response:[p. 9] We agree with the reviewer that short-term deflation events are not necessarily related to eruptive cycles but may be associated with the rate of magma supply. As shown by Chadwick et al. (2021), the supply rate has been waning since the 2015 eruption. The slow inflation may allow for the local movement of magma in the multiple reservoirs in the shallow crust. We have carefully checked the data from bottom pressure recorders (BPRs) in Chadwick et al. (2021) for the June 2018 short-term deflation event (see the following figure). The BPR on the western caldera wall, i.e., MJ03B (purple line), shows greater vertical displacement than those on the eastern walls, MJ03D (brown line) and MJ03E (green line). This may provide indirect evidence that this deflation event may have been caused by movement of magma beneath the western wall.

[Figure redacted]

3.20: 186-188. The lack of a magma pathway is consistent with magma not erupting along the western wall, but it is not an explanation of why there is no pathway.

Response:[p. 9] We agree with the review that the absence of a magma pathway and the lack of eruptions along the western wall do not provide an explanation for why there is no pathway. Through analysis of our high-resolution FWI models and previous seismicity studies, we observed that the outward-dipping fault beneath the western wall terminates at the notch between the MMR and WMR. In addition, there is no low-velocity stripping anomaly below the western volcano shoulder, as observed on the eastern shoulder. Considering these observations, we infer that there may not have magma pathways beneath the western wall.

3.21: 189-191. This is an odd choice of citations and prior work does not predict a broad magma body -indeed quite the opposite. At the EPR Dunn et al. (2000) report a width that “is narrow in the lower crust (5-7 km)” at 2-6 km depth and given the limits of travel time tomographic resolution, it presumably could be even narrower. Indeed, in a more recent paper Dunn (2022) estimates a width of 4.5 km at shallower depths broadening somewhat in the lower crust. Arnoux et al. (2019) also conclude the magma chamber is narrow in the lower crust along the Endeavour segment of the Juan de Fuca Ridge. The model in this manuscript only extends to 2.5 km bsf so it not at all clear how wide the low velocity zone is at greater depth. The inversions in this manuscript have higher resolution and thus resolve structure beneath the MMR that not been resolved previously but it is not clear that if the image extended to greater depth, it wouldnt be anything other than very consistent with the prior travel time tomography studies.

Response:[p. 9] We have thoroughly reviewed the references of Dunn et al. (2000), Dunn (2022), and Arnoux et al. (2019). In the original manuscript, we primarily referenced older papers, which may not have been the most appropriate. We have cited more recent works in the updated manuscript and revised this sentence for accuracy.

3.22: 201-204. Where else other than the upper mantle can the melt come from? What is purpose of this sentence.

Response:[p. 9] Eilon and Abers (2017) revealed a region 50 km wide and 150 km deep with a 2% melt fraction beneath the Axial segment. Here we cite their work to explain the magma source in the mantle, which contributes to the formation of a compressive magmatic system below Axial Seamount.

3.23: 221-227. This is interesting. 228-229. As noted above a 2-D image cannot show this.

Response:[p. 10] We agree with the reviewer on this point. The SMR lies to the south of the MMR, and the J48 survey line does not intersect the center of the SMR. Based on the current 2D profile, we cannot infer the connection between the MMR and SMR in actual 3D configuration. The description in the original manuscript may not be accurate. We have revised these sentences to clarify that the SMR may not connect to the MMR at shallow depths on the J48 survey line profile, and we cannot rule out intermittent connection zones between them that lie outside of the image plane.

REVIEWERS' COMMENTS

Reviewer #1 (Remarks to the Author):

Review of revised manuscript by William Chadwick

This is my second review of this manuscript, which has been substantially revised. I find that the revisions have addressed most of my concerns about the original manuscript and that the paper is much improved. However, it still needs some minor revision to respond to suggested edits and comments in the accompanying annotated manuscript.

Specific comments (keyed to lines in the annotated manuscript):

Line 2 (Title): Consider making the title more informative by adding something like this to the end: "..., based on full waveform inversion of seismic data"

Line 13 (Abstract): I think it's important to make clear that you are not using data from the whole seismic survey. Just one 2D line. I omitted "short-term deflation events" from the end of the Abstract, because they are a very limited and specific part of the observations at Axial, and this sentence is better as a "big picture" statement.

Line 18: Could add a hotspot reference here, such as:

Chadwick, J., M. Perfit, I. Ridley, I. Jonasson, G. Kamenov, W. W. Chadwick, Jr., R. Embley, P. Le Roux, and M. Smith (2005), Magmatic effects of the Cobb Hotspot on the Juan de Fuca Ridge, *Journal of Geophysical Research: Solid Earth*, 110, B03101, doi:10.1029/2003JB002767.

Line 32: Pre- and co-eruption would be 2014-2015 (not 2015-2016), but this statement really applies to all the seismicity, so I omitted that qualifier.

Lines 34-36: Good statement of motivation.

Lines 45-50: I omitted this because geodetic data isn't used to constrain the melt fraction.

Line 67 (Figure 1): In Figure 1b, it is actually difficult to see all the overlays because of the 3D topography. Consider making this a 2D map instead. I think it would make it much easier to see all the information.

Lines 67-83: This is a much improved description of methods.

Lines 87-88: To me, it would be better to leave out this comparison with Santorini, since it is a silicic system, and its relevance is questionable.

Lines 111-112: Is a "than" missing in this sentence? I think you are trying to say that the Carbotte imagery shows a wider conduit than in this study, right? Need to clarify wording here. Don't need phrase at the end of the sentence.

Line 113: What does "high melt-related viscosity" mean here? Does this imply relatively high crystal mush (vs. melt) content? If so, it would be clearer to use that terminology.

Lines 126-128: There already was a 3D MCS survey at Axial in 2019, led by Adrien Arnulf. Is that what you are referring to here? If so, this sentence should be re-written to clarify that. Or this sentence could be omitted.

Line 144: You don't need this first sentence.

Lines 211-213: This sentence seems a bit tangential to me and unnecessary. It isn't integral to the rest of the paragraph. Omit?

Lines 228-230: Again, this first sentence is unnecessary and could be omitted.

Lines 230-233: The evidence for the eastern side of the caldera being the source of most recent lava flows is not so much from the distribution of different lava flow types, it's that the eruptive fissures are clearly along the eastern edge of the caldera, and this is reflected in the relatively shallower bathymetry where the south rift zone enters the caldera from the south.

Lines 243-244: The wording of this sentence is unclear. Omit, since it is not necessary.

Line 249: I feel strongly that a sentence like this needs to be added here, to acknowledge the limitations of what you can say from just one 2D seismic line: "One limitation of this study is that it is limited to only one 2D seismic line across the summit of Axial Seamount. Additional insights might be gained by applying the analysis used here to other 2D lines collected during the 2002 MCS survey. This would test some of our ideas about how magma and partial melt is distributed in the shallow crust and the connections between them. For example, ..."

Reviewer #2 (Remarks to the Author):

Reviewer 2. My original review was that the paper be published after minor revision to address issues related to the quantification of melt fraction, with the strong suggestion that the authors include an "exceedance plot" for better quantifying uncertainty. The authors have added such a plot and use it effectively to address the issues that I raise. Additionally, after reading the complete rebuttal document and spot-checking the revised manuscript, I find that the authors to a very-reasonable job addressing the other reviewers' criticisms. Consequently, my opinion is that this paper is in excellent shape and may now be published.

Reviewer #3 (Remarks to the Author):

The authors have diligently but cautiously revised the manuscript and responded to the reviewers' comments, and it is certainly in a form that is publishable. As noted in my first review, the inversions are technically impressive, but in my view the revised manuscript still lacks a clear discernment of what is new and what just repeats previous ideas with a higher resolution model. There is nothing very exciting in the abstract which provides melt fractions that confirm existing ideas about the magmatic structure of Axial and mid-ocean ridge volcanos in general and notes a conduit connecting the magma chamber to eruptive fissures where we know one must exist. Only the last sentence alludes to something new. To my mind, an exciting new discovery is the western magma reservoir (WMR), which prior authors seems to overlook, and its link to the eruptive cycle and asymmetry of the volcano. I

suggested that the reviewers look for geodetic evidence to support their idea that magma escaped to the WMR from the main magma reservoir (MMR) during deflation events. The authors found this and presented it in their response to the reviewers, but it has not made its way into the manuscript. I am not sure why not.

On 3.18 (their numbering) they cite Hefner et al as evidence of less deformation on the west wall in the 2015 eruption. Hefner took the measurements of couple deformation on the faults from a seismic catalog (the figure in the response to the reviewers) as a constraint and then inverted them for a deflation model. That is not evidence that this was the actual slip on the fault.

The statement in the revised manuscript on line 251 “But we cannot rule out intermittent connection between the MMR and SMR that is located outside of the image plane” is still misleading because the authors cannot rule out a “permanent” connection based on a 2-D profile.

William Wilcock

Dear reviewers:

This is our response to the reviews of manuscript, entitled “Asymmetric magma plumbing beneath Axial Seamount based on full-waveform inversion of seismic data”. Comments from editors and reviewers are in red italics and our response is in black normal type. Page numbers of the modification corresponded to the comments are at the beginning of each response. A revised version with tracked changes is uploaded for supporting review, in which each response is marked at the margin. All figure numbers refer to the resubmitted manuscript, unless noted otherwise. Bibliographic references can be found in the manuscript.

I. Responses for Reviewer-1

1.1: Line 2 (Title): Consider making the title more informative by adding something like this to the end: “..., based on full waveform inversion of seismic data”

Response:[Title] We have changed the title to “Asymmetric magma plumbing system beneath Axial Seamount based on full-waveform inversion of seismic data”.

1.2: Line 13 (Abstract): I think it’s important to make clear that you are not using data from the whole seismic survey. Just one 2D line. I omitted “short-term deflation events” from the end of the Abstract, because they are a very limited and specific part of the observations at Axial, and this sentence is better as a “big picture” statement.

Response:[Abstract] We have clarified that the full-waveform inversion is for a seismic 2D line and deleted the description of “short-term deflation events” in the updated abstract.

1.3: Line 18: Could add a hotspot reference here, such as: Chadwick, J., M. Perfit, I. Ridley, I. Jonasson, G. Kamenov, W. W. Chadwick, Jr., R. Embley, P. Le Roux, and M. Smith (2005), Magmatic effects of the Cobb Hotspot on the Juan de Fuca Ridge, Journal of Geophysical Research: Solid Earth, 110, B03101, doi:10.1029/2003JB002767.

Response:[p. 2] We have added this reference in the updated manuscript.

1.4: Line 32: Pre- and co-eruption would be 2014-2015 (not 2015-2016), but this statement really applies to all the seismicity, so I omitted that qualifier.

Response:[p. 2] We agree with the reviewer about this point and have revised this sentence in the updated manuscript.

1.5: Lines 45-50: I omitted this because geodetic data isn't used to constrain the melt fraction.

Response:[p. 3] We have deleted the description about geodetic data in this sentence.

1.6: Line 67 (Figure 1): In Figure 1b, it is actually difficult to see all the overlays because of the 3D topography. Consider making this a 2D map instead. I think it would make it much easier to see all the information.

Response:[Fig. 1b] We have replotted Fig. 1b by adjusting the view angle and azimuth to make all features clear.

1.7: Lines 87-88: To me, it would be better to leave out this comparison with Santorini, since it is a silicic system, and its relevance is questionable.

Response:[p. 4] We have delete the description about the comparison with Santorini volcano.

1.8: Lines 111-112: Is a "than" missing in this sentence? I think you are trying to say that the Carbotte imagery shows a wider conduit than in this study, right? Need to clarify wording here. Don't need phrase at the end of the sentence.

Response:[p. 5] Yes. We have added "than" in the this sentence to make it clear.

1.9: Line 113: What does "high melt-related viscosity" mean here? Does this imply relatively high crystal mush (vs. melt) content? If so, it would be clearer to use that terminology.

Response:[p. 5] Yes. High crystal mush produces strong seismic attenuation due to the viscosity. We have revised this sentence according to the reviewer's suggestion.

1.10: Lines 126-128: There already was a 3D MCS survey at Axial in 2019, led by Adrien Arnulf. Is that what you are referring to here? If so, this sentence should be re-written to clarify that. Or this sentence could be omitted.

Response: We have deleted this sentence to avoid confusion.

1.11: Line 144: You don't need this first sentence.

Response: We have deleted this sentence in the updated manuscript.

1.12: Lines 211-213: This sentence seems a bit tangential to me and unnecessary. It isn't integral to the rest of the paragraph. Omit?

Response: We have deleted this sentence according to the reviewers' comment.

1.13: Lines 228-230: Again, this first sentence is unnecessary and could be omitted.

Response: We have deleted this sentence to make it more compact in the updated manuscript.

1.14: Lines 230-233: The evidence for the eastern side of the caldera being the source of most recent lava flows is not so much from the distribution of different lava flow types, it's that the eruptive fissures are clearly along the eastern edge of the caldera, and this is reflected in the relatively shallower bathymetry where the south rift zone enters the caldera from the south.

Response:[p. 10] We agree with the reviewer's about this point. These part has been revised according to reviewer's annotation in the updated manuscript.

1.15: Lines 243-244: The wording of this sentence is unclear. Omit, since it is not necessary.

Response: We have deleted this sentence according to the reviewers' suggestion.

1.16: Line 249: I feel strongly that a sentence like this needs to be added here, to acknowledge the limitations of what you can say from just one 2D seismic line: "One limitation of this study is that it is limited to only one 2D seismic line across the summit of Axial Seamount. Additional insights might be gained by applying the analysis used here to other 2D lines collected during the 2002 MCS survey. This would test some of our ideas about how magma and partial melt is distributed in the shallow crust and the connections between them. For example, "

Response:[p. 10, 11] We have added a couple of sentences to clarify the limitation of the 2D line analysis in the updated manuscript according to the reviewer's suggestion.

II. Responses for Reviewer 2

2.1: My original review was that the paper be published after minor revision to address issues related to the quantification of melt fraction, with the strong suggestion that the authors include an “exceedance plot” for better quantifying uncertainty. The authors have added such a plot and use it effectively to address the issues that I raise. Additionally, after reading the complete rebuttal document and spot-checking the revised manuscript, I find that the authors to a very-reasonable job addressing the other reviewers’ criticisms. Consequently, my opinion is that this paper is in excellent shape and may now be published.

Response: Thanks a lot for your suggestions. We have further revised the manuscript according to the other reviewers’ suggestions and comments.

III. Responses for Reviewer 3

3.1: The authors have diligently but cautiously revised the manuscript and responded to the reviewers’ comments, and it is certainly in a form that is publishable. As noted in my first review, the inversions are technically impressive, but in my view the revised manuscript still lacks a clear discernment of what is new and what just repeats previous ideas with a higher resolution model. There is nothing very exciting in the abstract which provides melt fractions that confirm existing ideas about the magmatic structure of Axial and mid-ocean ridge volcano in general and notes a conduit connecting the magma chamber to eruptive fissures where we know one must exist. Only the last sentence alludes to something new. To my mind, an exciting new discovery is the western magma reservoir (WMR), which prior authors seems to overlook, and its link to the eruptive cycle and asymmetry of the volcano. I suggested that the authors look for geodetic evidence to support their idea that magma escaped to the WMR from the main magma reservoir (MMR) during deflation events. The authors found this and presented it in their response to the reviewers, but it has not made its way into the manuscript. I am not sure why not.

Response:[p. 1, 3, 9] We have revised the abstract according to the reviewers’ suggestions and emphasized the newly discovered western magma reservoir in the abstract and main text. As for the geodetic evidence, we have added more description in the main text of the updated manuscript.

3.2: On 3.18 (their numbering) they cite Hefner et al as evidence of less deformation on the west wall in the 2015 eruption. Hefner took the measurements of couple deformation on the faults from a seismic catalog (the figure in the response to the reviewers) as a constraint and then inverted them for a deflation model. That is not evidence that this was the actual slip on the fault.

Response: We have carefully reviewed Hefner’s paper (2020) and fully agree with the reviewer on this point. The inverted slips from a deflation model might not represent the actual slips on the faults, but rather they may reflect the relative slip relationship of the faults beneath the western and eastern caldera walls.

3.3: The statement in the revised manuscript on line 251 “But we cannot rule out intermittent connection between the MMR and SMR that is located outside of the image plane” is still misleading because the authors cannot rule out a “permanent” connection based on a 2-D profile.

Response:[p. 11] In the updated manuscript, we have added the limitation of the 2D analysis according to the first reviewers’ suggestion, and also revised this sentence to avoid misleading.